# Sharp Gap-Dependent Variance-Aware Regret Bounds for Tabular MDPs

**Shulun Chen**[*]
Tsinghua University
chensl22@mails.tsinghua.edu.cn

**Runlong Zhou**
University of Washington
vectorzh@cs.washington.edu

**Zihan Zhang**
HKUST
zihanz@ust.hk

**Maryam Fazel**
University of Washington
mfazel@uw.edu

**Simon S. Du**
University of Washington
ssdu@cs.washington.edu

## Abstract

We consider gap-dependent regret bounds for episodic MDPs. We show that the Monotonic Value Propagation (MVP) algorithm (Zhang et al. [2024]) achieves a variance-aware gap-dependent regret bound of

$$\tilde{O}\left(\left(\sum_{\Delta_h(s,a)>0}\frac{H^2\log K\wedge\mathtt{Var}^{\mathrm{c}}_{\max}}{\Delta_h(s,a)}+\sum_{\Delta_h(s,a)=0}\frac{H^2\wedge\mathtt{Var}^{\mathrm{c}}_{\max}}{\Delta_{\min}}+SAH^4(S\vee H)\right)\log K\right),$$

where $H$ is the planning horizon, $S$ is the number of states, $A$ is the number of actions, $K$ is the number of episodes, and $\tilde{O}$ hides $\mathrm{poly}\log(S,A,H,1/\Delta_{\min},1/\delta)$ terms. Here, $\Delta_h(s,a) = V_h^*(a) - Q_h^*(s,a)$ represents the suboptimality gap and $\Delta_{\min} := \min_{\Delta_h(s,a)>0}\Delta_h(s,a)$. The term $\mathtt{Var}^{\mathrm{c}}_{\max}$ denotes the maximum conditional total variance, calculated as the maximum over all $(\pi, h, s)$ tuples of the expected total variance under policy $\pi$ conditioned on trajectories visiting state $s$ at step $h$. $\mathtt{Var}^{\mathrm{c}}_{\max}$ characterizes the maximum randomness encountered when learning any $(h, s)$ pair. Our result stems from a novel analysis of the weighted sum of the suboptimality gap and can be potentially adapted for other algorithms. To complement the study, we establish a lower bound of

$$\Omega\left(\sum_{\Delta_h(s,a)>0}\frac{H^2\wedge\mathtt{Var}^{\mathrm{c}}_{\max}}{\Delta_h(s,a)}\cdot\log K\right),$$

demonstrating the necessity of dependence on $\mathtt{Var}^{\mathrm{c}}_{\max}$ even when the maximum unconditional total variance (without conditioning on $(h, s)$) approaches zero.

## 1 Introduction

Reinforcement learning (RL, Sutton et al. [1998]) is an interactive decision-making problem where an agent gains information from an unknown environment through taking actions, with the goal

---

[*]Work done while Shulun Chen was visiting the University of Washington.

39th Conference on Neural Information Processing Systems (NeurIPS 2025).

of maximizing the total reward. RL has a wide range of applications, such as robotics and control [Lillicrap et al., 2015], games [Silver et al., 2016], finance [Nevmyvaka et al., 2006], healthcare [Liu et al., 2017], and recommendation systems [Chen et al., 2019].

The most canonical setting in RL is episodic learning in tabular Markov decision processes (MDPs), where the agent interacts with the MDP for $K$ episodes, each episode allowing exactly $H$ steps taken. Under this setting, we choose *cumulative regret* as the performance criteria, which should scale sublinearly with $K$ to indicate that the agent is making progress by shortening the performance difference between the policy $\pi^k$ played in episode $k$ and the optimal policy $\pi^*$. Most work [Azar et al., 2017, Jin et al., 2018, Dann et al., 2019, Zhang et al., 2020, 2021a] in this topic focused on *minimax regret* that is the worst-case guarantee for the algorithms over all the MDPs. Typically, these minimax regret bounds have main order terms scaling with $\sqrt{K}$.

The MDPs in practice often enjoy benign structures, so the above-mentioned algorithms may perform far better than their worst-case guarantees. Consequently, *problem-dependent* regret bounds are of great interest. *Variance-dependent* regret bounds [Talebi and Maillard, 2018, Zanette and Brunskill, 2019, Zhou et al., 2023, Zhang et al., 2024] are informative when the MDP is near-deterministic. This type of regret bounds have main order terms scaling with $\sqrt{\mathtt{Var} \cdot K}$ where $\mathtt{Var}$ is a symbol for some variance quantity (might be different across different works). For deterministic MDPs and MDPs such that $V_h^*(s) = V_h^*(s')$ for any $h, s, s'$, $\mathtt{Var} = 0$.

Meanwhile, *gap-dependent* regret bounds [Simchowitz and Jamieson, 2019, Yang et al., 2021, Dann et al., 2021, Xu et al., 2021, Zheng et al., 2024] are especially favored when for every $h, s$, the optimal value $V_h^*(s)$ is better than other suboptimal values $Q_h^*(s, a)$ by a margin. Formally, let $\Delta_h(s, a) := V_h^*(s) - Q_h^*(s, a)$ and $\Delta_{\min} := \min\{\Delta_h(s, a) \mid (h, s, a) \in [H] \times \mathcal{S} \times \mathcal{A}, \Delta_h(s, a) > 0\}$, then a typical gap-dependent regret bound is

$$\widetilde{O}\left(\left(\sum_{(h,s,a) \in \mathcal{Z}_{\mathrm{sub}}} \frac{1}{\Delta_h(s, a)} + \frac{|\mathcal{Z}_{\mathrm{opt}}|}{\Delta_{\min}} + \mathsf{poly}(H, S, A)\right) \mathsf{poly}(H) \cdot \log K\right), \qquad (1)$$

where $\mathcal{Z}_{\mathrm{sub}}$ is the set of all suboptimal $(h, s, a)$ tuples, $\mathcal{Z}_{\mathrm{opt}}$[2] is the set of all optimal $(h, s, a)$ tuples, and $\widetilde{O}$ hides $\mathsf{poly} \log(S, A, H, 1/\Delta_{\min}, 1/\delta)$ terms. When $K$ is large enough, gap-dependent regrets grow much slower than minimax and variance-dependent (when $\mathtt{Var} > 0$) regrets.

A natural yet fundamental question about problem-dependent regrets is:

> *What is the tightest problem-dependent regret while considering both variance and gap?*

If such a regret outperforms variance-only-dependent and gap-only-dependent regrets *asymptotically* (as $T \to \infty$) while also being nearly minimax optimal, it is actually ***best-of-three-worlds***!

To address the above problem, there are two factors that can be improved in previous gap-dependent regrets. First is the dependence on variance quantities. Only Simchowitz and Jamieson [2019], Zheng et al. [2024] contain variance-dependent terms in their gap-dependent regrets, while their variance quantities are defined as the *maximum per-step* variance, $\mathbb{Q}^* \leqslant H^2$. This quantity is first defined in Zanette and Brunskill [2019], and all of them use $H\mathbb{Q}^*$ as an *almost-sure upper bound* on variances. This upper bound can be substantially larger than an *expected total* variance (such as Definitions 5 and 6 in Zhou et al. [2023]). From this side, a tighter dependence on an expected total variance can improve the regret.

Second is the dependence on $H$. Specifically, when compared under the *time-inhomogeneous* setting, the $\mathsf{poly}(H)$ factors in Equation (1) are $H^3$, $H^6$, $H^5$, and $H^5$ in Simchowitz and Jamieson [2019], Yang et al. [2021], Xu et al. [2021], Zheng et al. [2024], respectively. Simchowitz and Jamieson [2019] provides a lower bound of $\Omega\left(\sum_{s,a} H^2/\Delta_1(s, a)\right)$, which indicates the chance of shaving out extra $H$ dependence.

**Our contributions.** We analyze the gap-dependent regret of the Monotonic Value Propagation (MVP, Zhang et al. [2024] version) algorithm, which is a model-based algorithm already proven to be near-optimal in the sense of minimax and variance-only-dependent regrets. After careful analysis,

---

[2]Xu et al. [2021] used a more fine-grained notion named $\mathcal{Z}_{\mathrm{mul}}$ instead.

we show that the gap-dependent regret depends on a variance quantity $\mathtt{Var}^c_{\max} \leqslant H\mathbb{Q}^*$, and the worst-case dependency on $H$ is $H^2$. ***We improve the above-mentioned two factors simultaneously.*** Formally, with probability at least $1 - \delta$, the regret in $K$ episodes by MVP is bounded as

$$\widetilde{O}\left(\left(\sum_{(h,s,a)\in\mathcal{Z}_{\mathrm{sub}}} \frac{H^2 \log K \wedge \mathtt{Var}^c_{\max}}{\Delta_h(s,a)} + \frac{(H^2 \wedge \mathtt{Var}^c_{\max})|\mathcal{Z}_{\mathrm{opt}}|}{\Delta_{\min}} + SAH^4(S \vee H)\right)\log K\right). \tag{2}$$

To the best of our knowledge, we are the first to incorporate a *tighter* variance quantity into gap-dependent regrets, and the worst-case dependency of $H^2$ in gap-dependent terms is also the state-of-the-art (see Table 1).

To complement our upper bound, we provide a lower bound (see Theorem 3) of

$$\Omega\left(\sum_{(h,s,a)\in\mathcal{Z}_{\mathrm{sub}}} \frac{H^2 \wedge \mathtt{Var}^c_{\max}}{\Delta_h(s,a)} \cdot \log K\right).$$

With this lower bound, we show that the first term in the upper bound (2) is tight (modulo log terms). This implies that (i) It is necessary to introduce the conditional total variance (see Definition 2) to derive a variance-aware gap-dependent bound. In comparison, the unconditional total variance (see Definition 1) is sufficient for variance-aware minimax bounds (e.g., Zhou et al. [2023]); (ii) When the first term in (2) dominates, the order of $H$ cannot be improved.

**Technical novelty.**   We propose a new variance metric to describe the upper bound of regret in gap-dependent MDPs. Our version of variance metric considers the conditional total variance to allow for some states with small visiting probability to accumulate a large regret over the whole training progress.

To derive a tighter regret bound using our new metric, we utilize a novel analysis which reweighs the suboptimality gaps. Our approach does not require the clipping and recursion method in Simchowitz and Jamieson [2019] for the main bound; instead, we directly prove that a certain weight sum over all suboptimality gaps times the visitation counts is bounded by a lower-order term of visitation counts, and establish a congregated upper bound of all visitation counts. We believe our approach is novel and reveals fundamental facts about suboptimality gaps.

We also propose a more refined version of clipping for optimal actions. Our version of clipping utilizes the new conditional variance metric while also providing an $O(H^2)$ worst case bound for $\Delta_{\min}$-dependent terms.

Finally, we prove that the $\Delta_h(s,a)$ terms in our upper bound match the lower bound modulo log factors. The construction is based on a reduction to Bernoulli bandits. A key insight is that low-frequency states, though often neglected in deriving minimax regret bounds, can still contribute substantially to regret in gap-dependent bounds.

**Paper overview.**   In Section 2, we introduce previous research about gap-dependent regret bound. In Section 3, we list the basic concepts of MDPs and define the conditional variance. In Section 4, we describe the MVP algorithm and provide a proof sketch of the gap-dependent regret upper bound. We conclude our paper in Section 5 with a matching lower bound.

## 2   Related works

**Gap-dependent regrets and sample complexities.**   Research on gap-dependent regrets originates from multi-armed bandits, which are special MDPs with $H = S = 1$. Auer et al. [2002] showed a $\sum_{a\in\mathcal{Z}_{\mathrm{sub}}} \log K/\Delta(a)$ type regret when running an UCB algorithm on MABs. Bubeck et al. [2012] proposed algorithms achieving a $\sum_{a\in\mathcal{Z}_{\mathrm{sub}}}(\Delta(a) + \log(1/\varepsilon)/\Delta(a))$ *bounded* regret given knowledge of the maximum reward $\max_a r(a)$ as well as a lower bound $\varepsilon > 0$ of $\Delta$.

Aside from the works studying finite-horizon tabular MDPs mentioned in Section 1, there is a line of work under the setting of gap-dependent regrets for infinite-horizon tabular MDPs [Auer and Ortner,

| Algorithm | Gap-dependent Regret | Variance-dependent | Minimax Optimal |
|---|---|---|---|
| StrongEuler [Simchowitz and Jamieson, 2019] | $\widetilde{O}(((\sum_{h,s,a} H\mathbb{Q}^*/(\Delta_h(s,a) \vee \Delta_{\min}) + SAH^4(S \vee H)) \cdot \log K)$ | Yes ($H\mathbb{Q}^*$) | **Yes** ($\tilde{O}(\sqrt{H^3SAK})$) |
| | $\widetilde{O}((\sum_{(h,s,a)\in\mathcal{Z}_{\text{sub}}} H^3/\Delta_h(s,a) + |\mathcal{Z}_{\text{opt}}|H^3/\Delta_{\min} + SAH^4(S \vee H)) \log K)$ | No | |
| Q-learning (UCB-H) [Yang et al., 2021] | $\widetilde{O}(H^6SA/\Delta_{\min} \cdot \log K)$ | No | No ($\tilde{O}(\sqrt{H^5SAK})$) [Jin et al., 2018] |
| AMB [Xu et al., 2021] | $\widetilde{O}((\sum_{(h,s,a)\in\mathcal{Z}_{\text{sub}}} H^5/\Delta_h(s,a) + |\mathcal{Z}_{\text{mul}}|H^5/\Delta_{\min}) \log K + SAH^2)$ | No | Not Provided |
| UCB-Advantage [Zheng et al., 2024] | $\widetilde{O}((H\mathbb{Q}^* + H)H^2SA/\Delta_{\min} \cdot \log K + S^2AH^9 \cdot \log^2 K)$ | Yes ($H\mathbb{Q}^*$) | **Yes** ($\tilde{O}(\sqrt{H^3SAK})$) [Zhang et al., 2020] |
| Q-EarlySettled-Advantage [Zheng et al., 2024] | $\widetilde{O}((H\mathbb{Q}^* + H^2)H^2SA/\Delta_{\min} \cdot \log K + SAH^7 \cdot \log^2 K)$ | Yes ($H\mathbb{Q}^*$) | **Yes** ($\tilde{O}(\sqrt{H^3SAK})$) [Li et al., 2021] |
| MVP This work | $\widetilde{O}((\sum_{(h,s,a)\in\mathcal{Z}_{\text{sub}}}(H^2 \log K \wedge \text{Var}_{\max}^c)/\Delta_h(s,a) + |\mathcal{Z}_{\text{opt}}|(H^2 \wedge \text{Var}_{\max}^c)/\Delta_{\min} + SAH^4(S \vee H)) \log K)$ | **Yes** ($H^2 \wedge \text{Var}_{\max}^c$) | **Yes** ($\tilde{O}(\sqrt{H^3SAK})$) [Zhang et al., 2024] |
| Lower Bound This work | $\Omega((\sum_{(h,s,a)\in\mathcal{Z}_{\text{sub}}}(H^2 \wedge \text{Var}_{\max}^c)/\Delta_h(s,a) \cdot \log K)$ | - | - |

Table 1: Comparison between different algorithms and their gap-dependent regrets for *time-inhomogeneous* MDPs. The result in Simchowitz and Jamieson [2019] is scaled accordingly as it originally studied *time-homogeneous* MDPs. **Variance-dependence:** whether the gap-dependent regret is also variance-dependent. $\text{Var}_{\max}^c \leqslant H\mathbb{Q}^*$, so dependence on $H^2 \wedge \text{Var}_{\max}^c$ is tighter. **Minimax Optimal:** whether the analyzed algorithm achieves a $\widetilde{O}(\sqrt{H^3SAK})$ (main order) minimax regret. Xu et al. [2021] did not provide such a guarantee.

2006, Tewari and Bartlett, 2007, Auer et al., 2008, Ok et al., 2018], while in these works, the gaps are usually defined as the difference between policies instead of actions. Recently, gap-dependent regrets have been studied for risk-sensitive RL [Fei and Xu, 2022], linear/general function classes [He et al., 2021, Papini et al., 2021, Velegkas et al., 2022], and Markov games [Dou et al., 2022].

Gap-dependent sample complexities under online [Jonsson et al., 2020, Marjani and Proutiere, 2020, Al Marjani et al., 2021, Wagenmaker et al., 2022b, Tirinzoni et al., 2022, Wagenmaker and Jamieson, 2022, Tirinzoni et al., 2023] and offline [Wang et al., 2022, Nguyen-Tang et al., 2023] RL setting are also widely studied.

**Minimax optimal regrets.** Under the setting of time-inhomogeneous MDPs, algorithms achieving a high-probability regret upper bound of $\widetilde{O}(\sqrt{H^3SAK})$ are *(nearly) minimax optimal*. There have been many works with this guarantee while optimizing the lower order terms: Azar et al. [2017], Osband and Van Roy [2017], Zanette and Brunskill [2019], Simchowitz and Jamieson [2019], Zhang and Ji [2019], Zhang et al. [2020, 2021a], Ménard et al. [2021], Li et al. [2021], Xiong et al. [2022], Zhou et al. [2023], Zhang et al. [2024]. Notably, Zhang et al. [2024] derived the tightest $\widetilde{O}(\sqrt{H^3SAK} \wedge HK)$ regret up to logarithm factors.

**Variance-dependent regrets.** Talebi and Maillard [2018] studied variance-dependent regrets for infinite horizon learning under strong assumptions on ergodicity of the MDPs. Zanette and Brunskill [2019] defined and incorporated the maximum per-step conditional variance, $\mathbb{Q}^*$, and first proved a $\widetilde{O}(\sqrt{H\mathbb{Q}^* \cdot SAK})$ regret for the finite-horizon setting. Zhou et al. [2023], Zhang et al. [2024] proved regrets depending on expected total variances (see our Definition 2 for one of their quantities) that are more fine-grained than the coarse $H\mathbb{Q}^*$ upper bound. Variance-dependent regrets have also been studied for bandits [Zhang et al., 2021b, Zhou et al., 2021, Kim et al., 2022, Dai et al., 2022].

**Other problem-dependent regrets.** Under infinite-horizon setting, Bartlett and Tewari [2012], Fruit et al. [2018] studied regrets depending on the span of the optimal value function. There are works studying first-order regrets, whose main order terms depend on value functions: Jin et al. [2020], Wagenmaker et al. [2022a], Huang et al. [2023].

## 3 Preliminaries

**Notations.** For any event $\mathcal{E}$, let $\mathbf{1}\{\mathcal{E}\}$ be the indicator function of $\mathcal{E}$. For any set $\mathcal{X}$, we use $\Delta^{\mathcal{X}}$ to denote the probability simplex over $\mathcal{X}$. For any positive integer $n$, we denote $[n] := \{1, 2, \ldots, n\}$. $\tilde{O}, \tilde{\Omega}, \lesssim$ hide $\text{poly} \log(S, A, H, 1/\Delta_{\min}, 1/\delta)$ factors.

**Finite-horizon MDPs and trajectories.** A finite-horizon MDP is described by a tuple $M = (\mathcal{S}, \mathcal{A}, H, P, R, \mu)$. $\mathcal{S}$ is the finite state space with size $S$ and $\mathcal{A}$ is the finite action space with size $A$. $H$ is the planning horizon. For any $(s, a, h) \in \mathcal{S} \times \mathcal{A} \times [H]$, $P_{s,a,h} \in \Delta^{\mathcal{S}}$ is the transition

function and $R_{s,a,h} \in \Delta^{[0,H]}$ is the reward distribution with mean $r_h : \mathcal{S} \times \mathcal{A} \to [0, H]$. $\mu \in \Delta^{\mathcal{S}}$ is the initial state distribution. A trajectory $\{s_1, a_1, r'_1, s_2, a_2, r'_2, \cdots, s_H, a_H, r'_H\}$ is sampled with $s_1 \sim \mu, s_{h+1} \sim P_{s_h,a_h,h}, r'_h \sim R_{s_h,a_h,h}$ where $a_h$ can be chosen arbitrarily.

Unlike most common settings, we relax the standard assumption that $R_{s,a,h} \in \Delta^{[0,1]}$ (uniformly bounded reward) and instead assume a bounded total reward setting (Assumption 1). Problems under this setting can contain a spike in reward and are therefore harder than standard problems.

**Assumption 1** (Bounded total reward). *We assume that $\sum_{h=1}^{H} r'_h \leqslant H$ for any possible trajectory.*

**Policies.** A history-independent deterministic policy $\pi$ chooses an action based on the current state and time step. Formally, $\pi = \{\pi_h\}_{h \in [H]}$ where $\pi_h : \mathcal{S} \to \mathcal{A}$ maps a state to an action. Any trajectory sampled by $\pi$ satisfies $a_h = \pi_h(s_h)$. For any random variable $X$ related to a trajectory, we denote $\mathbb{E}^\pi[X]$ and $\mathbb{V}^\pi[X]$ as the expectation and variance of $X$ when the trajectory is sampled under $\pi$.

**Value functions and $Q$-functions.** Given $\pi$, we define its value function and $Q$-function as

$$V_h^\pi(s) := \mathbb{E}^\pi \left[ \sum_{t=h}^{H} r_t \,\middle|\, s_h = s \right], \quad Q_h^\pi(s, a) := \mathbb{E}^\pi \left[ \sum_{t=h}^{H} r_t \,\middle|\, (s_h, a_h) = (s, a) \right].$$

It is easy to verify that $Q_h^\pi(s, a) = r_h(s, a) + P_{s,a,h}V_{h+1}^\pi$. We define $V_0^\pi := \mathbb{E}^{s \sim \mu}[V_1^\pi(s)]$ as the expected total reward when executing policy $\pi$.

**Learning objective.** Episodic RL on MDPs proceeds for a total of $K$ episodes. At the beginning of episode $k$, the learner chooses a policy $\pi^k$ and uses it to sample a trajectory.

We aim to maximize $V_0^\pi$. Using dynamic programming, we can find a policy $\pi^*$ maximizing all $Q_h^\pi(s, a)$ simultaneously, and we denote $V^* := V^{\pi^*}, Q^* := Q^{\pi^*}$.

Performance is evaluated by the cumulative regret:

$$\text{Regret}(K) := \sum_{k=1}^{K} \left( V_0^* - V_0^{\pi^k} \right).$$

**Gap quantities.** The suboptimality gap is defined as follows:

$$\Delta_h(s, a) := V_h^*(s) - Q_h^*(s, a).$$

The sets of optimal and suboptimal actions are defined as

$$\mathcal{Z}_{\text{opt}} = \{(s, a, h) \in \mathcal{S} \times \mathcal{A} \times [H] : \Delta_h(s, a) = 0\}, \quad \mathcal{Z}_{\text{sub}} = \mathcal{S} \times \mathcal{A} \times [H] \backslash \mathcal{Z}_{\text{opt}}.$$

The minimum gap $\Delta_{\min} = \min_{(s,a,h) \in \mathcal{Z}_{\text{sub}}} \Delta_h(s, a)$ is the smallest positive gap. WLOG, we only consider MDPs with nonempty $\mathcal{Z}_{\text{sub}}$.

**Variance quantities.** The variance at each $(s, a, h)$ tuple [Zanette and Brunskill, 2019, Simchowitz and Jamieson, 2019] is defined as

$$\texttt{Var}_h^*(s, a) := \mathbb{V}^{r \sim R_{s,a,h}, s' \sim P_{s,a,h}} [r + V_{h+1}^*(s')].$$

The maximum per-step conditional variance is defined as $\mathbb{Q}^* := \max_{h,s,a} \texttt{Var}_h^*(s, a)$. Previous works including Zheng et al. [2024] use $H\mathbb{Q}^*$ which could be as large as $H^3$ in their variance-dependent terms.

The maximum *unconditional* total variance has been introduced in prior works [Zhou et al., 2023, Zhang et al., 2024] when studying variance-dependent regret bounds for MDPs.

**Definition 1** (Maximum unconditional total variance).

$$\texttt{Var}_{\max} := \max_\pi \mathbb{E}^\pi \left[ \sum_{h=1}^{H} \texttt{Var}_h^*(s_h, a_h) \right].$$

These works showed that $\mathtt{Var}_{\max} \lesssim \min\{H\mathbb{Q}^*, H^2\}$ and incorporated it in the main order terms of variance-only-dependent regrets for better results. However, as we will discuss in Theorem 3, variance-aware gap-dependent regrets *must* scale with separate variance quantities for each $(s, h)$ pair, even for those hard to visit. Thus, the quantity should be conditioned on $(s, h)$. We propose the following quantity as the maximum *conditional* total variance:

**Definition 2** (Maximum conditional total variance)**.**

$$\mathtt{Var}_{\max}^{\mathrm{c}} := \max_{\pi, s, h} \mathbb{E}^{\pi} \left[ \sum_{h'=1}^{H} \mathtt{Var}_{h'}^*(s_{h'}, a_{h'}) \,\middle|\, s_h = s \right].$$

**Remark 1.** The maximum conditional total variance is novel in literature, as in variance-only-dependent works, $\mathtt{Var}_{\max}$ is a better quantity, while in previous variance-aware gap-dependent works, researchers did not develop better approaches other than bounding total variance by $H\mathbb{Q}^*$. By definition, $\mathtt{Var}_{\max}^{\mathrm{c}} \leqslant H\mathbb{Q}^*$, and our final results will scale with $\min\{\mathtt{Var}_{\max}^{\mathrm{c}}, H^2\}$ after careful analysis, which can improve the dependency on $H$ by one order.

# 4 Main Results

## 4.1 Algorithm Overview: MVP

Monotonic Value Propagation (MVP, Appendix B) is a representative [Zhang et al., 2021a, Zhou et al., 2023, Zhang et al., 2024] model-based optimistic algorithm which maintains upper bounds of $V^*$ and $Q^*$, namely $V^k$ and $Q^k$, in each episode. The rollout policy $\pi^k$ picks the action that maximizes $Q_h^k(s, \cdot)$ at each step and updates the upper bounds using Bellman equation with empirical estimates of reward and transitions:

$$Q_h(s, a) \leftarrow \hat{r}_h(s, a) + \mathbb{E}^{s \sim \hat{P}_{s,a,h}} V_{h+1}(s, a) + b_h(s, a), \quad V_h(s) \leftarrow \max_a Q_h(s, a).$$

Here $b_h(s, a)$ is a bonus term ensuring that $Q_h, V_h$ are upper bounds of $Q_h^*, V_h^*$ ("optimism") with high probability. For the proof of optimism, interested readers can refer to Zhang et al. [2021a].

## 4.2 Gap-dependent Upper Bound

Now, we present the main result of this work – a gap-dependent regret upper bound ensured by MVP. For the formal version, please refer to Appendix C.

**Theorem 2** (Gap-dependent upper bound)**.** *There exists universal constants $c_1, c_2, c_3$ such that, for any MDP instance, any episode number $K$, and $\delta > 0$, MVP (Algorithm 1) attains the following regret bound with probability at least $1 - \delta$:*

$$\mathrm{Regret}(K) \lesssim \left( \sum_{(h,s,a) \in \mathcal{Z}_{\mathrm{sub}}} \frac{H^2 \log K \wedge \mathtt{Var}_{\max}^{\mathrm{c}}}{\Delta_h(s, a)} + \frac{(H^2 \wedge \mathtt{Var}_{\max}^{\mathrm{c}})|\mathcal{Z}_{\mathrm{opt}}|}{\Delta_{\min}} + SAH^4(S \vee H) \right) \log K.$$

**Remark 2.** This bound contains a new notion of maximum conditional total variance (Definition 2). Since this definition requires us to condition on any possible state, $\mathtt{Var}_{\max}^{\mathrm{c}}$ can be as large as $\Theta(H^3)$. However, $\mathtt{Var}_{\max}^{\mathrm{c}}$ is bounded by $H \max_{s,a,h}\{\mathtt{Var}_h^*(s, a, h)\}$, so this term is still no worse than previous $O(H\mathbb{Q}^*)$ bounds. Furthermore, there is a *sufficient* condition to make $\mathtt{Var}_{\max}^{\mathrm{c}} = O(H^2 \log(1/\delta))$: for any policy $\pi$ and $(s, a, h) \in \mathcal{S} \times \mathcal{A} \times [H]$, the state-action pair $(s, a)$ is not reachable at step $h$ if we sample the trajectory under $\pi$, or it is visited with probability at least $\delta$. We can even generalize this concept to exclude the states that are difficult to reach from the definition of $\mathtt{Var}_{\max}^{\mathrm{c}}$. We omit this approach for simplicity.

The $H^2$ term in $H^2 \wedge \mathtt{Var}_{\max}^{\mathrm{c}}$ is derived by conditioning on the event where all trajectories have bounded total variance to avoid the dependence on $\mathtt{Var}_{\max}^{\mathrm{c}}$ when it is large.

Our lower bound (Theorem 3) shows that $\mathtt{Var}_{\max}^{\mathrm{c}}$ cannot be replaced by $\mathtt{Var}_{\max}$ (Definition 2), a quantity used by previous variance-only-dependent works. Intuitively, $\mathtt{Var}_{\max}$ can be very small as long as all states with large variance have small visiting probability, but those states can accumulate a total regret of order $\mathtt{Var}_h^*(s, a)/\Delta_h(s, a)$ that cannot be bounded by $\mathtt{Var}_{\max}$. The leading term matches with the lower bound modulo $\log$ factors. Furthermore, Simchowitz and Jamieson [2019] has shown that a $\Delta_{\min}$-dependent term is unavoidable for UCB-based algorithms. Our coefficient of the $\Delta_{\min}$ term is also improved to worst case $O(H^2)$, better than previous worst-case factors of $H\mathbb{Q}^*$ [Simchowitz and Jamieson, 2019] and $H^3\mathbb{Q}^*$ [Zheng et al., 2024].

## 4.3 Proof Sketch

We present the high-level ideas in the proof for Theorem 2 here, deferring the details to Appendix C. We assume the optimistic condition holds (see Lemma 9).

**Regret decomposition.** Following Simchowitz and Jamieson [2019], we define

$$E_h^k(s,a) := Q_h^k(s,a) - (R_h(s,a) + \mathbb{E}^{s' \sim P_{s,a,h}}[V_{h+1}^k(s')]) \tag{3}$$

as the surplus at $(s,a,h,k) \in \mathcal{S} \times \mathcal{A} \times [H] \times [K]$. Standard analysis show the regret bound

$$\text{Regret}(K) \lesssim \mathbb{E}\left[\sum_{k=1}^{K}\sum_{h=1}^{H} E_h^k(s,a)\right].$$

**Analyzing gaps and surpluses.** Suppose that the algorithm takes action $a$ at state $s$ at episode $k$, stage $h$. By optimism, $Q_h^k(s,a)$ must be at least $V_h^*(s) = Q_h^*(s,a) + \Delta_h(s,a)$, so we have

$$E_h^k(s,a) + \mathbb{E}^{s' \sim P_{s,a,h}}[V_{h+1}^k(s') - V_{h+1}^*(s')] \geqslant \Delta_h(s,a).$$

By recursively expanding the $V$ term, we have

$$\Delta_h(s,a) \leqslant \mathbb{E}^{\pi^k}\left[\sum_{h'=h}^{H} E_h^k(s,a)\middle|(s_{h'},a_{h'}) = (s,a)\right]. \tag{4}$$

That is, if the expectation of future surpluses is small, then the algorithm will avoid actions with large suboptimality gap.

The analysis of $E_h^k$ shows that (see Lemma 16)

$$E_h^k(s,a) \lesssim \sqrt{\frac{\text{Var}_h^*(s,a)\iota}{n_h^k(s,a)}} + \underbrace{\sum_{h' \geqslant h} \mathbb{E}^{\pi^k}\left[\frac{SH\iota}{n_h^k(s_h,a_h)}\middle|s_h = s\right]}_{\text{low order terms}}.$$

We consider the restrictions of $n_h^k(s,a)$ when $\Delta_h(s,a) > 0$. If the lower bound Equation (4) wrote $\Delta_h(s,a) \lesssim E_h^k(s,a)$, then $n_h^k(s,a) \lesssim \text{Var}_h^*(s,a)\iota/\Delta_h(s,a)^2$, which would directly provide a regret bound. However, Equation (4) contains the sum all future surpluses, so we cannot directly apply this method.

We will circumvent this problem by adding Equation (4) over all $k$ and $h$. The summation of the left-hand side is $\sum_{s,a,h} \Delta_h(s,a)n_h^k(s,a)$, while the summation of the right-hand side can be shown as approximately (low-order terms discarded) $H\sum_{s,a,h}\sqrt{\text{Var}_h^*(s,a)n_h^k(s,a)\iota}$.

This inequality has the form $\sum_{s,a,h} u_{s,a,h}n_h(s,a) \lesssim \sum_{s,a,h} v_{s,a,h}\sqrt{n_h(s,a)}$ for some non-negative coefficients $u_{s,a,h}, v_{s,a,h}$. It entails upper bounds of $n_h(s,a)$, and if we proceed with the calculations, we will recover the bound

$$\text{Regret}(K) \lesssim \sum_{s,a,h} \frac{H\text{Var}_h^*(s,a)\iota}{\Delta_h(s,a)} + (\text{some low-order terms})$$

in Simchowitz and Jamieson [2019] while avoiding complex calculations. In the latter steps, we will refine this method for a tighter bound.

**Generalized weighted sum of suboptimality gaps.** Intuitively, the previous bound is not balanced as $\text{Var}_h^*(s,a) = \Omega(H^2)$ only happens for a small portion of $(s,a,h)$. In contrast, the summation of Equation (4) contains enough degrees of freedom for us to utilize it for a better bound. Let $w_h(s,a)$ be any set of nonnegative weights. Then the weighted sum of Equation (4) writes

$$\sum_{s,a,h} w_h(s,a)\Delta_h(s,a)n_h^K(s,a) \lesssim \sum_{h=1}^{H}\sum_{k=1}^{K} w_h(s_h^k,a_h^k)\sum_{h'=h}^{H} \mathbb{E}[E_{h'}^k(s_{h'}^k,a_{h'}^k)|\mathcal{F}_{k-1,h}], \tag{5}$$

where $\mathcal{F}_{k-1,h}$ is the $\sigma$-field generated by first $k-1$ episodes and the first $h$ states in the $k$-th trajectory. We will choose $w_h(s,a)$ carefully to balance the contribution of each term.

**Bounding weighted sum of surpluses.** The right-hand side of Equation (5) needs to be manipulated carefully. Rewriting the summation order, the leading term of Equation (5) becomes

$$\sum_{s',a'} \sum_{h'=1}^{H} \sum_{k=1}^{K} \sqrt{\frac{\text{Var}^*_{h'}(s',a')\iota}{n^k_{h'}(s',a')}} \sum_{h=1}^{h'} w_h(s^k_h, a^k_h) \mathbb{E}[\mathbf{1}\{(s^k_{h'}, a^k_{h'}) = (s',a')\}|\mathcal{F}_{k-1,h}].$$

We will apply certain probability inequalities to approximate $\mathbb{E}[\mathbf{1}\{(s^k_{h'}, a^k_{h'}) = (s',a')\}|\mathcal{F}_{k-1,h}]$ with $\mathbf{1}\{(s^k_{h'}, a^k_{h'}) = (s',a')\}$ (approximation error omitted). Then, the innermost sum over $h$ contains only $w_h(s^k_h, a^k_h)$, which can be bounded by $\bar{W} = H^2\iota \wedge \text{Var}^c_{\max}$ if we pick $w^k_h(s,a) = \text{Var}^*_h(s,a)$. Now, the sum over $k$ is

$$\sum_{n=1}^{n^K_{h'}(s',a')} \sqrt{\frac{\text{Var}^*_{h'}(s',a')\iota}{n}} = O\left(\sqrt{\text{Var}^*_{h'}(s',a')n^K_{h'}(s',a')\iota}\right),$$

so by Equation (5),

$$\sum_{s,a,h} Var^*_h(s,a)\Delta_h(s,a)n^K_h(s,a) \lesssim \bar{W} \underbrace{\sum_{s,a,h} \sqrt{\text{Var}^*_h(s,a)n^K_h(s,a)\iota}}_{:=R}. \tag{6}$$

**End of proof.** With similar (and simpler) arguments above, we have

$$\text{Regret}(K) \lesssim R,$$

where again, we ignore the lower-order terms.

We apply the Cauchy-Schwartz inequality to Equation (6) and get

$$\bar{W}R \cdot \left(\sum_{s,a} \sum_{h=1}^{H} \frac{\iota}{\Delta_h(s,a)}\right) \gtrsim \left(\sum_{s,a,h} \sqrt{\text{Var}^*_h(s,a)n^K_h(s,a)\iota}\right)^2 = R^2,$$

so we conclude that

$$\text{Regret}(K) \lesssim R \lessgtr \sum_{s,a,h} \frac{\bar{W}\iota}{\Delta_h(s,a)}.$$

## 5  Gap-Dependent Lower Bound

In this section, we will prove the following gap-dependent regret lower bound. It shows a separation between $\text{Var}^c_{\max}$ and $\text{Var}_{\max}$, as well as the necessity of $\text{Var}^c_{\max}$ in gap-dependent regrets.

**Theorem 3.** *[Gap-dependent lower bound (informal)] Fix $S, A, H$ and the target conditional variance $L \in [1, H^2]$. Given a set of $SAH$ suboptimality gaps $\{\Delta_i\}$, assume that all non-zero gaps are sufficiently small. For any algorithm, there always exists an MDP with gaps equal to $\Theta(\Delta_i)$, $\text{Var}^c_{\max} = \Theta(L)$ but $\text{Var}_{\max} = O(1)$, such that*

$$\text{Regret}(K) \geq \Omega\left(\sum_{i:\Delta_i>0} \frac{L}{\Delta_i} \cdot \log K\right).$$

**Proof sketch.** We sketch the proof as follows. For simplicity, we assume there are 4 states $\{\texttt{A}, \texttt{B}, \texttt{C}, \texttt{D}\}$ in each $h$-th layer. The dynamics of the four states are presented below.

- **A** : There is only one action at A, which transits the agent to A in the next layer with probability $1 - \frac{1}{LH}$, and B with probability $\frac{1}{LH}$. The reward is $0$ at A;
- **B** : There are $A$ actions at B. For each action $a$, the agent is transported to C with probability $\frac{1}{2} - \frac{\Delta(a_i)}{4\sqrt{L}}$ and D with probability $\frac{1}{2} + \frac{\Delta(a_i)}{4\sqrt{L}}$;
- **C** : This state is a terminal state with reward $\sqrt{L}$;

- `D` : This state is a terminal state with reward 0.

In this instance, the learner makes a decision only at state `B` for each layer $h$, and state `A` has variance $O(\frac{1}{LH} \cdot (\sqrt{L})^2) = O(H^{-1})$ and state `B` has variance $\Theta(L)$, showing $\mathtt{Var}^{\mathrm{c}}_{\max} = \Theta(L)$. For any strategy $\pi$, it visits `B` with probability $1 - (1 - \frac{1}{LH})^H = O(L^{-1})$, So

$$\mathtt{Var}_{\max} \leqslant H \cdot O(H^{-1}) + LO(L^{-1}) = O(1).$$

Clearly, the decision problem at state `B` and layer $h$ could be viewed as a Bernoulli bandit problem. The expected visiting count at state `B` and layer $h$ is $\Theta(K/L)$. Let $\mathrm{Regret}_{h,\mathtt{B}}(K)$ be the regret by taking suboptimal actions at `B` and the $h$-th layer. Consequently, applying the classical lower bound on regret for Bernoulli bandits yields:

$$\lim_{K \to \infty} \frac{\mathrm{Regret}_{h,\mathtt{B}}(K)}{\log(K/L)} \geqslant \Omega\left( \sum_a \frac{L}{\Delta_h(\mathtt{B}, a)} \right).$$

Thus,

$$\mathrm{Regret}(K) \geqslant \sum_{h=1}^{H} \mathrm{Regret}_{h,\mathtt{B}}(K) \geqslant \Omega\left( \sum_{h,a} \frac{L}{\Delta_h(\mathtt{B}, a)} \cdot \log K \right).$$

for sufficiently large $K$.

**Discussion.** This example shows a separation between unconditional variance $\mathtt{Var}^{\mathrm{c}}_{\max}$ and conditional variance $\mathtt{Var}_{\max}$. Even if $\mathtt{Var}_{\max} = O(1)$, there can still be a regret lower bound of order $\Theta(H^2)$. In this view, our introduction to $\mathtt{Var}^{\mathrm{c}}_{\max}$ is essential in proving gap-dependent regret bounds.

We also observe that the second term $\frac{(H^2 \wedge \mathtt{Var}^{\mathrm{c}}_{\max})|\mathcal{Z}_{\mathrm{opt}}|}{\Delta_{\min}}$ in our upper bound (2) is not yet matched by this lower bound. This could pose a significant challenge for existing optimistic algorithms, as they typically explore all potentially optimal actions, resulting in additional surplus terms. We refer the readers to Appendix D the full proof of Theorem 3.

## 6 Conclusion

In this paper, we study gap-dependent regret bounds for episodic MDPs and demonstrate that the Monotonic Value Propagation (MVP) algorithm Zhang et al. [2024] achieves a tighter upper bound compared to previous works from the aspects of tighter dependence on a better variance notion, as well as reduced order of $H$. Our analysis centers around a careful bound of the weighted sum of suboptimality gaps. Along the way, we introduce a new notion of *maximum conditional total variance* and provide a lower bound to establish its necessity as well as the tightness of the $\frac{1}{\Delta_h(s,a)}$ term.

We also acknowledge some limitations. First, the $\frac{(H^2 \wedge \mathtt{Var}^{\mathrm{c}}_{\max})|\mathcal{Z}_{\mathrm{opt}}|}{\Delta_{\min}}$ term in our upper bound does not match the lower bound of $\frac{S}{\Delta_{\min}}$ in Theorem 2.3 of Simchowitz and Jamieson [2019]. Improving either the upper bound or lower bound will help advancing the understanding of gap-dependent regrets. Second, we only apply our new techniques to tabular MDPs. For future work, we believe our analysis can be adapted to other problem settings (e.g., linear MDPs [Wagenmaker and Jamieson, 2022] and MDPs with general function approximation) to derive tighter gap-dependent regret bounds.

## Acknowledgement

SSD acknowledges the support of NSF DMS 2134106, NSF CCF 2212261, NSF IIS 2143493, NSF IIS 2229881, Alfred P. Sloan Research Fellowship, and Schmidt Sciences AI 2050 Fellowship. RZ and MF acknowledge the support of NSF TRIPODS II DMS-2023166. The work of MF was supported in part by awards NSF CCF 2212261 and NSF CCF 2312775.

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

# A  Notations and Technical Lemmas

## A.1  Notations

We list notations in Tables 2 to 4.

| | |
|---|---|
| $\mathcal{S}, S = \|\mathcal{S}\|$ | State space and its size |
| $\mathcal{A}, A = \|\mathcal{A}\|$ | Action space and its size |
| $H$ | Horizon |
| $K$ | Learning episodes |
| $s, s'$ | States in $\mathcal{S}$ |
| $a, a'$ | Actions in $\mathcal{A}$ |
| $h, h', h^*$ | Horizon numbers |
| $k, k'$ | Indices of learning episode |
| $P_{s,a,h}$ | Transition probability |
| $R_{s,a,h}$ | Distribution of rewards |
| $\mu$ | Distribution of beginning state |
| $r_h(s, a)$ | Expected reward |
| $\pi$ | Policy |
| $\pi_h(s)$ | Action that policy $\pi$ takes at state $s$, step $h$ |
| $V_h^\pi(s), V_h^*(s)$ | $V$-function of policy $\pi$ and of optimal policy, respectively |
| $Q_h^\pi(s, a), Q_h^*(s, a)$ | $Q$-function of policy $\pi$ and of optimal policy, respectively |
| $\mathrm{Var}_h^*(s, a)$ | Variance at state $s$, action $a$, and step $h$ |
| $\mathrm{Var}_{\max}$ | Maximum unconditional variance |
| $\mathrm{Var}_{\max}^c$ | Maximum conditional variance |
| $\Delta_h(s, a)$ | Suboptimality gap |
| $\Delta_{\min}$ | Minimal nonzero suboptimality gap |
| $\mathcal{Z}_{\mathrm{sub}}$ | Set of suboptimal actions |
| $\mathcal{Z}_{\mathrm{opt}}$ | Set of optimal actions |

Table 2: Parameters of MDP

| | |
|---|---|
| $s_h^k, a_h^k, r_h^k$ | States, actions, and rewards observed in the $k$-th episode |
| $V_h^k(s)$ | $V_h$ of the algorithm before the $k$-th episode |
| $Q_h^k(s, a)$ | $Q_h$ of the algorithm before the $k$-th episode |
| $\hat{r}_h^k(s, a)$ | Estimation of $r_h(s, a)$ before the $k$-th episode |
| $\hat{\sigma}_h^k(s, a)$ | Estimation of $\sigma_h(s, a)$ before the $k$-th episode |
| $\hat{P}_{s,a,h}^k$ | Estimation of $P_{s,a,h}$ before the $k$-th episode |
| $\hat{n}_h^k(s, a)$ | Visitation count at $(s, a, h)$ before the $k$-th episode |
| $b_h^k(s, a)$ | Bonus term in the $k$-th episode |
| $\pi^k$ | The policy at the $k$-th episode |

Table 3: Values used in the algorithm

## A.2  Technical Lemmas

**Lemma 1** (Bennett's inequality, Theorem 3 in Maurer and Pontil [2009]). *Let $X_1, X_2, \cdots, X_n$ be i.i.d. random variables with values $[0, a](a > 0)$ and let $\delta > 0$. Then,*

$$\mathbb{P}\left[\left|\mathbb{E}[X_1] - \frac{1}{n}\sum_{i=1}^n X_i\right| > \sqrt{\frac{2\mathbb{V}[X_1]\log(2/\delta)}{n}} + \frac{a\log(2/\delta)}{n}\right] < \delta.$$

**Lemma 2** (Freedman's inequality, Lemma 10 in Zhang et al. [2020]). *Let $(X_n)_{n \geqslant 1}$ be a martingale difference sequence (i.e., $\mathbb{E}[X_n|\mathcal{F}_{n-1}] = 0$ for all $n \geqslant 1$, where $\mathcal{F}_k = \sigma(X_1, X_2, \cdots, X_k)$) such that $|X_n| \leqslant a$ for some $a > 0$ and for all $n \geqslant 1$. Let $V_n = \sum_{k=1}^n \mathbb{E}[X_k^2|\mathcal{F}_{k-1}]$ for $n \geqslant 0$. Then, for any positive integer $n$, and any $\varepsilon, \delta > 0$, we have*

$$\mathbb{P}\left[\left|\sum_{i=1}^n X_i\right| \geqslant 2\sqrt{V_n\log(1/\delta)} + 2\sqrt{\varepsilon\log(1/\delta)} + 2a\log(1/\delta)\right] \leqslant 2(na^2\varepsilon^{-1} + 1)\delta.$$

| | |
|---|---|
| $[n]$ | Set $\{1, 2, \cdots, n\}$ |
| $\Delta^B$ | Set of distribution functions over set $B$ |
| $x \wedge y$ | $\min\{x, y\}$ |
| $x \vee y$ | $\max\{x, y\}$ |
| $\mathbf{1}\{\varphi\}$ | Indicator function of $\varphi$, i.e. 1 if $\varphi$ is true and 0 otherwise |
| $\mathrm{clip}\,[a\|\varepsilon]$ | $a\mathbf{1}\{a \geqslant \varepsilon\}$ |
| $\delta$ | Acceptable error probability |
| $E_h^k(s,a)$ | Surplus; $Q_h^k(s,a) - r_h(s,a) - \mathbb{E}^{s' \sim P_{s,a,h}}[V_{h+1}^k(s')]$ |
| $\bar{E}_h^k(s,a)$ | Clipped surplus |
| $\iota$ | $\log(SAHK/\delta)$ |
| $w_h(s,a)$ | Weights used in analysis |
| $\bar{W}$ | $160H^2 \log(4K(H+1)/\delta) \wedge \mathtt{Var}_{\max}^{\mathrm{c}}$ |
| Regret | Total regret |
| $\mathcal{F}_k$ | $\sigma$-field generated by the first $k-1$ episodes of the algorithm |
| $\mathcal{F}_{k,h}$ | $\sigma$-field generated by the first $k-1$ episodes and the first $h$ steps in the $k$-th episode |
| $\sum_s, \sum_a, \sum_{s,a}$ | $\sum_{s \in \mathcal{S}}, \sum_{a \in \mathcal{A}}, \sum_{s \in \mathcal{S}, a \in \mathcal{A}}$, respectively |
| $\mathbb{E}^{x \sim X}, \mathbb{V}^{x \sim X}$ | Expectation when $x$ is sampled from distribution $X$ |
| $\mathbb{P}^\pi, \mathbb{E}^\pi$ | Probability and expectation over a trajectory when following policy $\pi$ |

Table 4: Other notations

**Lemma 3** (Lemma 10 in Zhang et al. [2022])**.** *Let $X_1, X_2, \ldots$ be a sequence of random variables taking values in $[0, l]$. Define $\mathcal{F}_k = \sigma(X_1, X_2, \ldots, X_{k-1})$ and $Y_k = \mathbb{E}[X_k \mid \mathcal{F}_k]$ for $k \geqslant 1$. For any $\delta > 0$, we have that*

$$\mathbb{P}\left[\exists n, \sum_{k=1}^n X_k \geqslant 3\sum_{k=1}^n Y_k + l\ln(1/\delta)\right] \leqslant \delta,$$

$$\mathbb{P}\left[\exists n, \sum_{k=1}^n Y_k \geqslant 3\sum_{k=1}^n X_k + l\ln(1/\delta)\right] \leqslant \delta.$$

**Lemma 4** (Lemma F.5 in Simchowitz and Jamieson [2019])**.** *Let $X, Y$ be two random variables defined on the same probability space. Then*

$$|\sqrt{\mathbb{V}[X]} - \sqrt{\mathbb{V}[Y]}| \leqslant \sqrt{\mathbb{E}[(X-Y)^2]}.$$

**Lemma 5** (Lemma B.5 in Simchowitz and Jamieson [2019])**.** *Let $a_1, a_2, \cdots, a_m$ be a sequence of nonnegative reals and $\varepsilon > 0$. Then,*

$$\mathrm{clip}\left[\sum_{i=1}^m a_i \Big| \varepsilon\right] \leqslant 2\sum_{i=1}^m \mathrm{clip}\left[a_i \Big| \frac{\varepsilon}{2m}\right].$$

### A.3 Model errors

Our analysis will mostly be based on the success of following inequalities.

**Lemma 6** (Good events)**.** *Let $\iota = \log(SAHK/\delta)$. With probability at least $1 - 10\delta$, the following inequalities hold for all $s, a, s', h, k$:*

$$|\hat{r}_h^k(s,a) - r_h(s,a)| \leqslant \sqrt{\frac{2\mathbb{V}^{r' \sim R_{s,a,h}}[r']\iota}{n_h^k(s,a)}} + \frac{H\iota}{n_h^k(s,a)},$$

$$|\hat{P}_{s,a,h}^k(s') - P_{s,a,h}(s')| \leqslant \sqrt{\frac{2P_{s,a,h}(s')\iota}{n_h^k(s,a)}} + \frac{\iota}{n_h^k(s,a)},$$

$$|\mathbb{E}^{s' \sim \hat{P}_{s,a,h}^k}[V_{h+1}^*(s')] - \mathbb{E}^{s' \sim P_{s,a,h}^k}[V_{h+1}^*(s')]| \leqslant \sqrt{\frac{2\mathbb{V}^{s' \sim P_{s,a,h}}V_{h+1}^*(s')\iota}{n_h^k(s,a)}} + \frac{H\iota}{n_h^k(s,a)},$$

$$\sqrt{\mathbb{V}^{s'\sim\hat{P}^k_{s,a,h}}[V^*_{h+1}(s')]} - \sqrt{\mathbb{V}^{s'\sim P_{s,a,h}}[V^*_{h+1}(s')]} \leqslant H\sqrt{\frac{2\iota}{n^k_h(s,a)-1}}.$$

$$\sqrt{\hat{\sigma}^k_h(s,a)-(\hat{r}^k_h(s,a))^2} - \sqrt{\mathbb{V}^{r'\sim R_{s,a,h}}[r']} \leqslant H\sqrt{\frac{2\iota}{n^k_h(s,a)-1}}.$$

*Proof.* The first three inequalities can be derived from Theorem 1, Theorem 2. The last two inequalities are adapted from Theorem 10 in Maurer and Pontil [2009]. $\square$

**Lemma 7.** *Let $V$ be a function defined on $\mathcal{S}$. Conditioned on the success of Lemma 6,*

$$|\mathbb{E}^{s'\sim\hat{P}_{s,a,h}}[V(s')] - \mathbb{E}^{s'\sim P_{s,a,h}}[V(s')]| \leqslant \sqrt{\frac{2S\mathbb{E}^{s'\sim P_{s,a,h}}[V(s')^2]\iota}{n^k_h(s,a)}} + \frac{\max_{s\in\mathcal{S}}|V(s)|S\iota}{n^k_h(s,a)}.$$

*Proof.* Let $M = \max_{s\in\mathcal{S}}|V(s)|$. Then, by Lemma 6,

$$|\mathbb{E}^{s'\sim\hat{P}_{s,a,h}}[V(s')] - \mathbb{E}^{s'\sim P_{s,a,h}}[V(s')]|$$

$$= \left|\sum_{s'\in\mathcal{S}}(\hat{P}^k_{s,a,h}(s') - P_{s,a,h}(s'))V(s')\right|$$

$$\leqslant \sum_{s'\in\mathcal{S}}|V(s')|\left(\sqrt{\frac{2P_{s,a,h}(s')\iota}{n^k_h(s,a)}} + \frac{\iota}{n^k_h(s,a)}\right)$$

$$\leqslant \sqrt{\left(\sum_{s'\in\mathcal{S}}P_{s,a,h}(s')V(s')^2\right)\left(\sum_{s'\in\mathcal{S}}\frac{2\iota}{n^k_h(s,a)}\right)} + \frac{MS\iota}{n^k_h(s,a)}$$

$$= \sqrt{\mathbb{E}^{s'\sim P_{s,a,h}}[V(s')^2]\cdot\frac{2S\iota}{n^k_h(s,a)}} + \frac{MS\iota}{n^k_h(s,a)}.$$

$\square$

## A.4 Variance bounds

**Corollary 4.** *Let $\pi$ be any fixed policy. For any $h \in H$ and $s \in \mathcal{S}$, we have*

$$\mathbb{E}^\pi\left[\sum_{h'=h}^H \mathtt{Var}^*_h(s_{h'},a_{h'})\Bigg|s_h=s\right] \leqslant H^2.$$

*Proof.* Recall that $\mathbb{E}^{s'\sim P_{s,a,h}}[V^*_{h+1}(s')] = Q^*_h(s,a) - r_h(x,a) \leqslant V^*_h(s) - r_h(s,a)$, so

$$\mathbb{E}^\pi\left[\sum_{h'=h}^H \mathtt{Var}^*_h(s_{h'},a_{h'})\Bigg|s_h=s\right]$$

$$= \mathbb{E}^\pi\left[\sum_{h'=h}^H \mathbb{V}^{r'\sim R_{s_{h'},a_{h'},h'}}[r'] + \sum_{h=h'}^H \mathbb{V}^{s'\sim P_{s_{h'},a_{h'},h'}}[V^*_{h'+1}(s')]\Bigg|s_h=s\right]$$

$$\leqslant \mathbb{E}^\pi\left[\sum_{h'=h}^H (r'_{h'} - r_{h'}(s_{h'},a_{h'}))^2 + \sum_{h'=h}^H (V^*_{h'+1}(s_{h'+1}) - V^*_{h'}(s_{h'},a_{h'}) + r_{h'}(s_{h'}))^2\Bigg|s_h=s\right]$$

$$\leqslant \mathbb{E}^\pi\left[\left(\sum_{h'=h}^H (r'_{h'} + V^*_{h'+1}(s_{h'+1}) - V^*_{h'}(s_{h'}))\right)^2\Bigg|s_h=s\right] \tag{7}$$

$$\leqslant \mathbb{E}^\pi \left[ \left( \sum_{h=h'}^H r'_{h'} - V_h^*(s_h) \right)^2 \middle| s_h = s \right] \leqslant H^2,$$

where Equation (7) is because of independence and that

$$\mathbb{E}^{s' \sim P_{s_{h'}, a_{h'}, h}}[V_{h'+1}^*(s') - V_{h'}^*(s_{h'}, a_{h'}) + r_{h'}(s_{h'})] \leqslant 0 = \mathbb{E}^{r' \sim R_{s_{h'}, a_{h'}, h}}[r' - r_{h'}(s_{h'}, a_{h'})].$$

$\square$

**Lemma 8** (Lemma 42, Zhou et al. [2023]). *Let $\pi$ be any fixed policy. For any $\delta > 0$,*

$$\mathbb{P}^\pi \left[ \sum_{h'=h}^H \mathtt{Var}_h^*(s_{h'}, a_{h'}) \geqslant 160 H^2 \log(4(H+1)/\delta) \middle| s_h = s \right] \leqslant \delta.$$

*Proof.* We have

$$\sum_{h'=h}^H \mathtt{Var}_h^*(s_{h'}, a_{h'}) = \sum_{h'=h}^H \mathbb{V}^{r' \sim R_{s_{h'}, a_{h'}, h'}}[r'] + \sum_{h'=h}^H \mathbb{V}^{s' \sim P_{s_{h'}, a_{h'}, h'}}[V_{h'+1}^*(s')]$$

$$= \sum_{h'=h}^H \mathbb{E}^{s' \sim P_{s_{h'}, a_{h'}, h'}}[V_{h'+1}^*(s')^2] - \sum_{h'=h}^H (Q_{h'}^*(s_{h'}, a_{h'}) - r_{h'}(s_{h'}, a_{h'}))^2 + \sum_{h'=h}^H H r_{h'}(s_{h'}, a_{h'})$$

$$\leqslant \sum_{h'=h}^H (\mathbb{E}^{s' \sim P_{s_{h'}, a_{h'}, h'}}[V_{h'+1}^*(s')^2] - V_{h'+1}^*(s_{h'+1})^2)$$

$$+ \sum_{h'=h}^H (V_{h'}^*(s_{h'})^2 - (Q_{h'}^*(s_{h'}, a_{h'}) - r_{h'}(s_{h'}, a_{h'}))^2) + H^2$$

$$\leqslant 2\sqrt{2 \sum_{h'=h}^H \mathbb{V}^{s' \sim P_{s_{h'}, a_{h'}, h'}}[V_{h'+1}^*(s_{h'+1})^2] \log(1/\delta)} + 2\sqrt{H^4 \log(1/\delta)} + 2H^2 \log(1/\delta) \qquad (8)$$

$$+ 2H \sum_{h'=h}^H (V_{h'}^*(s_{h'}) - Q_{h'}^*(s_{h'}, a_{h'}) + r_{h'}(s_{h'}, a_{h'})) + H^2$$

$$\leqslant 4H \sqrt{2 \sum_{h'=h}^H \mathbb{V}^{s' \sim P_{s_{h'}, a_{h'}, h'}}[V_{h'+1}^*(s')^2]) \log(1/\delta)} + 5H^2 \log(1/\delta) + 2H \cdot V_h^*(s_h)$$

$$+ 4H \sqrt{2 \sum_{h'=h}^H \mathbb{V}^{s' \sim P_{s_{h'}, a_{h'}, h'}}[V_{h'+1}^*(s')^2]) \log(1/\delta)} + 4H\sqrt{H^2 \log(1/\delta)} + 4H^2 \log(1/\delta) \quad (9)$$

$$\leqslant 8H \sqrt{2 \sum_{h'=h}^H \mathtt{Var}_{h'}^*(s_{h'}, a_{h'})} + 15H^2 \log(1/\delta),$$

where Equations (8) and (9) holds with probability $1 - 2(H+1)\delta$ each by Lemma 2. Thus, by solving the quadratic equation,

$$\mathbb{P}^\pi \left[ \sum_{h'=h}^H \mathtt{Var}_h^*(s_{h'}, a_{h'}) \geqslant 160 H^2 \log(1/\delta) \middle| s_h = s \right] \leqslant 1 - 4(H+1)\delta.$$

The proof is finished with rescaling $\delta$. $\square$

# B  MVP Algorithm descrpition

This section provides a description of MVP algorithm (Algorithm 1) in detail[3].

---

[3]The original version of MVP contains a doubling mechanism to trigger updates of $V$ and $Q$ mainly to lower switching cost. Since switching cost is not central to gap-dependent analysis, we choose to update $V_h$ and $Q_h$ every episode for simplicity.

---

**Algorithm 1** Monotonic Value Propagation (MVP)

---

**Require:** MDP $\mathcal{M} = (\mathcal{S}, \mathcal{A}, H, P, R, \mu)$, learning episode number $K$, confidence parameter $\delta$, universal constants $c_1, c_2, c_3, \iota = \log(SAHK/\delta)$.

1: Initialize: For all $(s, a, s', h) \in \mathcal{S} \times \mathcal{A} \times \mathcal{S} \times [H+1]$, set $\theta_h(s,a), \kappa_h(s,a) \leftarrow 0, n_h(s,a,s') \leftarrow 0, n_h(s,a), Q_h(s,a), V_h(s) \leftarrow 0$.
2: **for** $k = 1, 2, \cdots, K$ **do**
3:     Construct policy $\pi^k$ such that $\pi_h^k(s) = \arg\max_a Q_h(s,a)$.
4:     Observe trajectory $s_1^k, a_1^k, r_1^k, s_2^k, a_2^k, r_2^k, \cdots, s_h^k, a_h^k, r_h^k$.
5:     **for** $h = H, H-1, \ldots, 1$ **do**
6:         $(s, a, s') \leftarrow (s_h^k, a_h^k, s_{h+1}^k)$
7:         Update $n_h(s,a,s') \leftarrow n_h(s,a,s') + 1, n_h(s,a) \leftarrow n_h(s,a) + 1, \theta_h(s,a) \leftarrow \theta_h(s,a) + r_h^k, \kappa_h(s,a) \leftarrow \kappa_h(s,a) + (r_h^k)^2$.
8:         $\hat{r}_h(s,a) = \frac{\theta_h(s,a)}{n_h(s,a)}$
9:         $\hat{\sigma}_h(s,a) = \frac{\kappa_h(s,a)}{n_h(s,a)}$
10:         $\hat{P}_h(s,a,s') = \frac{n_h(s,a,s')}{n_h(s,a)}$
11:         **for** $(s,a) \in \mathcal{S} \times \mathcal{A}$ **do**
12:             $b_h(s,a) \leftarrow c_1 \sqrt{\frac{\mathbb{V}^{s' \sim \hat{P}_{s,a,h}}[V_{h+1}(s')]\iota}{n_h(s,a) \vee 1}} + c_2 \sqrt{\frac{(\hat{\sigma}_h(s,a) - (\hat{r}_h(s,a))^2)\iota}{n_h(s,a) \vee 1}} + c_3 \frac{H\iota}{n_h(s,a) \vee 1}$
13:             $Q_h(s,a) \leftarrow \min\{\hat{r}_h(s,a) + \mathbb{E}^{s' \sim \hat{P}_{s,a,h}} V_{h+1} + b_h(s,a), H\}$
14:             $V_h(s) \leftarrow \max_a Q_h(s,a)$
15:         **end for**
16:     **end for**
17: **end for**

---

# C    Proof of main theorem

We begin by choosing the universal constants in the algorithm as $c_1 = c_2 = 2, c_3 = 10$.

## C.1    Clipping surpluses

Existing analysis of MDP already shows that $Q_h^k$ and $V_h^k$ are upper bounds of $Q_h^*$ and $V_h^*$ with high probability as expected:

**Lemma 9.** *With probability at least $1 - 4\delta$, for all $s, a, h, k \in \mathcal{S} \times \mathcal{A} \times [H] \times [K]$,*

$$Q_h^k(s,a) \geqslant Q_h^*(s,a), V_h^k(s) \geqslant V_h^*(s).$$

*Proof.* The proof is almost the same as Lemma 8 in Zhang et al. [2024] with necessary modifications for our constant choices. Since $c_3 = 10 \geqslant 4 = c_1^2$, the monotonic function can be constructed as

$$f_{P,n}(v) := \mathbb{E}^{s \sim P}[v(s)] + \max\left\{ 2\sqrt{\frac{\mathbb{V}^{s \sim P}[v(s)]\iota}{n}}, \frac{4H\iota}{n} \right\}.$$

$\square$

We define clipped surpluses as

$$\bar{E}_h^k(s_h, a_h) = \text{clip}\left[ E_h^k(s_h, a_h) \Big| c_4 \Delta_{\min} \max\left\{ \frac{\text{Var}_h^*(s,a)}{\min\{H^2, \text{Var}_{\max}^c\}} + \frac{1}{H} \right\} \right]. \qquad (10)$$

Also, we recursively define

$$\bar{Q}_{H+1}^k(s,a) = \bar{V}_{H+1}^k(s) = 0,$$

$$\bar{Q}_h^k(s,a) = r_h^k(s,a) + \bar{E}_h^k(s,a) + \mathbb{E}^{s' \sim P_{s,a,h}}[\bar{V}_{h+1}^k(s)], \bar{V}_h^k(s) = \max_a \bar{Q}_h^k(s,a)$$

for $h = H, H-1, \cdots, 1$, and $\bar{Q}_0^k = \bar{V}_0^k = \mathbb{E}^{s' \sim \mu}[\bar{V}_1^k(s')]$.

**Lemma 10.**
$$\bar{V}_h^k(s) \geqslant V_h^k(s) - \frac{\Delta_{\min}}{3}.$$

*Proof.* We have $\bar{E}_h^k(s,a) \geqslant E_h^k(s,a) - \frac{\Delta_{\min}\mathtt{Var}_h^*(s,a)}{6(H^2 \wedge \mathtt{Var}_{\max}^c)} - \frac{\Delta_{\min}}{6H}$ for any pair of $s, a, h$. Thus,

$$\bar{V}_h^k(s) - V_h^k(s)$$
$$=\mathbb{E}^{\pi^k}\left[\sum_{h'=h}^H \bar{E}_h^k(s_h, a_h)\bigg|s_h = s\right]$$
$$\geqslant\mathbb{E}^{\pi^k}\left[\sum_{h'=h}^H E_h^k(s_h, a_h)\bigg|s_h = s\right] - \frac{\Delta_{\min}}{6(H^2 \wedge \mathtt{Var}_{\max}^c)}\mathbb{E}^k\left[\sum_{h'=h}^H \mathtt{Var}_h^*(s_h, a_h)\bigg|s_h = s\right] - \sum_{h'=h}^H \frac{\Delta_{\min}}{6H}$$
$$\geqslant V_h^k(s) - V_h^k(s) - \frac{\Delta_{\min}}{3},$$

where the last line is due to Theorem 4 and definition of $\mathtt{Var}_{\max}^c$. $\qquad\square$

This lemma links the half-clipped values $\bar{V}_h^k$ with the optimal values $V_h^*$.

**Lemma 11.** *Conditioned on success of Lemma 9, for any state $s \in \mathcal{S}$ and $h \in [H]$,*

$$V_h^*(s) - V_h^k(s) \leqslant \frac{3}{2}(\bar{V}_h(s) - V_h^k(s)).$$

*Proof.* The first step in the proof is to recursively expand both sides at all states where an optimal action is taken. Specifically, we let

$$\mathcal{E}_{h*} = \{\pi_{h'}^k(s_{h'}) = a_{h'}, h' = h, h+1, \cdots, h^*\},$$

and $\mathcal{E}_{h*} - \mathcal{E}_{h*+1}$ as the set of the trajectories in $\mathcal{E}_{h*}$ but not in $\mathcal{E}_{h*+1}$ (that is, those trajectories where the first suboptimal action after the $h$-step is at the $(h^*+1)$-th step). Since trajectories are sampled with policy $\pi^k$, $\mathcal{E}_h$ is the set of all trajectories with $s_h = s$.

We hope to claim that

$$V_h^*(s) - V_h^k(s) = \sum_{h'=h+1}^{h^*} \mathbb{E}^{\pi^k}[\mathbf{1}\{\mathcal{E}_{h'-1} - \mathcal{E}_{h'}\}(V_{h'}^*(s_{h'}) - V_{h'}^{\pi^k}(s_{h'}))|s_h = s] \qquad (11)$$
$$+ \mathbb{E}^{\pi^k}[\mathbf{1}\{\mathcal{E}_{h*}\}(V_{h*+1}^*(s_{h*+1}) - V_{h*+1}^{\pi^k}(s_{h*+1})|s_h = s] \qquad (12)$$

and

$$\bar{V}_h^k(s) - V_h^k(s) \geqslant \sum_{h'=h+1}^{h^*} \mathbb{E}^{\pi^k}[\mathbf{1}\{\mathcal{E}_{h'-1} - \mathcal{E}_{h'}\}(\bar{V}_{h'}^k(s_{h'}) - V_{h'}^{\pi^k}(s_{h'}))|s_h = s] \qquad (13)$$
$$+ \mathbb{E}^{\pi^k}[\mathbf{1}\{\mathcal{E}_{h*}\}(\bar{V}_{h*+1}^k(s_{h*+1}) - V_{h*+1}^{\pi^k}(s_{h*+1}))|s_h = s]. \qquad (14)$$

These claims are proved by induction on $h^*$ and expanding the last term on event $\mathcal{E}_{h*+1}$. For Equation (11), we have

$$V_{h*}^*(s_{h*}) - V_{h*}^{\pi^k}(s_{h*})$$
$$=Q_{h*}^*(s_{h*}, \pi_{h*}^k(s_{h*})) - Q_{h*}^{\pi^k}(s_{h*}, \pi_{h*}^k(s_{h*}))$$
$$=\mathbb{E}^{s' \sim P_{s,\pi_{h*},k}}[V_{h*+1}^*(s') - V_{h*+1}^*(s')]$$

when the trajectory is in $\mathcal{E}_{h*+1}$, and for Equation (13), we have

$$\bar{V}_{h*}^k(s) - V_{h*}^{\pi^k}(s) = \bar{Q}_{h*}^k(s, \pi_{h*}^k(s)) - Q_{h*}^{\pi^k}(s, \pi_{h*}^k(s))$$
$$=\bar{E}_{h*}^k(s, a) + \mathbb{E}^{s' \sim P_{s,\pi_{h*}(s),k}}[\bar{V}_{h*+1}^k(s') - \bar{V}_{h*+1}^{\pi^k}(s')]$$

$$\geqslant \mathbb{E}^{s' \sim P_{s,\pi_{h^*}^k}(s),k}[\bar{V}_{h^*+1}^k(s') - \bar{V}_{h^*+1}^{\pi^k}(s')].$$

We will use Equation (11) and Equation (13) when $h^* = H$. In this case, the last lines are both zero, so it suffices to show that

$$\frac{3}{2}(\bar{V}_{h'}^k(s_{h'}) - V_{h'}^{\pi^k}(s_{h'})) \geqslant V_{h'}^*(s_{h'}) - V_{h'}^{\pi^k}(s_{h'})$$

on $\mathcal{E}_{h'-1} - \mathcal{E}_{h'}$. In fact, since the trajectory is sampled from $\pi^k$, and since $a_{h'}$ is suboptimal, we have that

$$\bar{V}_{h'}^k(s_{h'}) \geqslant V_h^k(s_{h'}) - \frac{\Delta_{\min}}{3} = Q_{h'}^k(s_{h'}, a_{h'}) - \frac{\Delta_{\min}}{3} \geqslant V_{h'}^*(s_{h'}) - \frac{\Delta_{h'}(s_{h'}, a_{h'})}{3} \geqslant \frac{2}{3}V_{h'}^*(s_{h'}),$$

where the last inequality is because $\Delta_{h'}(s_{h'}, a_{h'}) = V_{h'}^{\pi^k}(s_{h'}) - Q_{h'}^*(s_{h'}, a_{h'}) \leqslant V_{h'}^{\pi^k}(s_{h'})$. $\qquad\square$

**Lemma 12.** *Conditioned on success of Lemma 9, if $a = \pi_h^k(s)$, then*

$$\Delta_h(s, a) \leqslant \frac{3}{2} \sum_{h'=h}^H \mathbb{E}^{\pi^k}[\bar{E}_{h'}^k(s_{h'}, a_{h'})|(s_h, a_h)].$$

**Lemma 13.** *Conditioned on success of Lemma 9,*

$$V_0^* - V_0^{\pi^k} \leqslant \frac{3}{2} \sum_{h=1}^H \mathbb{E}^{\pi^k}[\bar{E}_h^k(s_h, a_h)].$$

*Proof.* We prove Lemmas 12 and 13 together. By recursively applying

$$\bar{V}_h^k(s) - V_h^k(s) = \bar{E}_h^k(s, \pi_h^k(s)) + \mathbb{E}^{s' \sim P_{s,\pi_h^k(s),h}}[\bar{V}_{h+1}^k(s) - V_{h+1}^{\pi^k}(s)],$$

we have

$$\bar{V}_h^k(s) - V_h^k(s) = \sum_{h'=h}^H \mathbb{E}^{\pi^k}\left[\bar{E}_{h'}^k(s_{h'}, a_{h'})|s_h = s\right].$$

Then we use Lemma 11 and

$$\Delta_h(s, a) = V_h^*(s) - Q_h^*(s, a) \leqslant V_h^*(s) - V_h^k(s), \quad V_0^* - V_0^{\pi^k} = \mathbb{E}^{\pi^k}[V_1^*(s_1) - V_1^{\pi^k}(s_1)],$$

for Lemmas 12 and 13, respectively. $\qquad\square$

### C.2 Estimating Surpluses

**Lemma 14.** *Conditioned on success of Lemma 6,*

$$b_h^k(s, a) \leqslant \frac{2}{H}\mathbb{E}^{s' \sim \hat{P}_{s,a,h}^k}[(V_{h+1}^k(s') - V_{h+1}^*(s'))^2] + 2\sqrt{\frac{2\mathtt{Var}_h^*(s,a)\iota}{n_h^k(s,a)}} + \frac{20H\iota}{n_h^k(s,a)}.$$

*Proof.* Recall that our choice of bonus in the algorithm is

$$b_h^k(s, a) = 2\sqrt{\frac{\mathbb{V}^{s' \sim \hat{P}_{s,a,h}^k}[V_{h+1}(s')]\iota}{n_h^k(s,a)}} + 2\sqrt{\frac{(\hat{\sigma}_h^k(s,a) - (\hat{r}_h^k(s,a))^2)\iota}{n_h^k(s,a)}} + \frac{10H\iota}{n_h^k(s,a)}.$$

Since the last term is at least $H$ if $n_h^k(s, a) = 1$, it suffices to consider $n_h^k(s, a) \geqslant 2$. The first term can be bounded using

$$\sqrt{\mathbb{V}^{s' \sim \hat{P}_{s,a,h}^k}[V_{h+1}^k(s')]} = \left(\sqrt{\mathbb{V}^{s' \sim \hat{P}_{s,a,h}^k}[V_{h+1}^k(s')]} - \sqrt{\mathbb{V}^{s' \sim \hat{P}_{s,a,h}^k}[V_{h+1}^*(s')]}\right)$$

$$+ \left(\sqrt{\mathbb{V}^{s' \sim \hat{P}_{s,a,h}^k}[V_{h+1}^*(s')]} - \sqrt{\mathbb{V}^{s' \sim P_{s,a,h}^k}[V_{h+1}^*(s')]}\right) + \sqrt{\mathbb{V}^{s' \sim P_{s,a,h}^k}[V_{h+1}^*(s')]}$$

$$\leqslant \sqrt{\mathbb{E}^{s'\sim \hat{P}^k_{s,a,h}}[(V^k_{h+1}(s') - V^*_{h+1}(s'))^2]} + H\sqrt{\frac{2\iota}{n^k_h(s,a) - 1}} + \sqrt{\mathbb{V}^{s'\sim P^k_{s,a,h}}[V^*_{h+1}(s')]}$$

by Lemmas 4 and 6, so

$$\sqrt{\frac{\mathbb{V}^{s'\sim \hat{P}^k_{s,a,h}}[V_{h+1}(s')]\iota}{n^k_h(s,a)}}$$

$$\leqslant \sqrt{\mathbb{E}^{s'\sim \hat{P}^k_{s,a,h}}[(V^k_{h+1}(s') - V^*_{h+1}(s'))^2] \cdot \frac{\iota}{n^k_h(s,a)} + \frac{2H\iota}{n^k_h(s,a)} + \sqrt{\frac{\mathbb{V}^{s'\sim P^k_{s,a,h}}[V^*_{h+1}(s')]\iota}{n^k_h(s,a)}}}$$

$$\leqslant \frac{1}{H}\mathbb{E}^{s'\sim \hat{P}^k_{s,a,h}}[(V^k_{h+1}(s') - V^*_{h+1}(s'))^2] + \sqrt{\frac{\mathbb{V}^{s'\sim P^k_{s,a,h}}[V^*_{h+1}(s')]\iota}{n^k_h(s,a)}} + \frac{3H\iota}{n^k_h(s,a)}.$$

The second term of $b^k_h(s,a)$ can easily be bounded by Lemma 6 as

$$\sqrt{\frac{(\hat{\sigma}^k_h(s,a) - (\hat{r}^k_h(s,a))^2)\iota}{n^k_h(s,a)}} \leqslant \sqrt{\frac{\mathbb{V}^{r'\sim R_{s,a,h}}[r']\iota}{n^k_h(s,a)}} + \frac{2H\iota}{n^k_h(s,a)}.$$

Thus

$$b^k_h(s,a) \leqslant \frac{2}{H}\mathbb{E}^{s'\sim \hat{P}^k_{s,a,h}}[(V^k_{h+1}(s') - V^*_{h+1}(s'))^2] + 2\sqrt{\frac{2\mathtt{Var}^*_h(s,a)\iota}{n^k_h(s,a)}} + \frac{20H\iota}{n^k_h(s,a)}.$$

$\square$

**Lemma 15.** *Conditioned on success of Lemma 6,*

$$V^k_h(s) - V^*_h(s) \leqslant \mathbb{E}^{\pi^k}\left[\sum_{h'=h}^{H} H \wedge 22H\sqrt{\frac{S\iota}{n^k_{h'}(s_{h'}, a_{h'})}}\,\bigg|\, s_h = s\right]$$

*Proof.* We begin by decomposing $V^k_h(s') - V^*_h(s')$ as follows:

$$V^k_h(s) - V^*_h(s) \leqslant V^k_h(s) - Q^*_h(s, \pi^k_h(s))$$
$$=\hat{r}^k_h(s,a) + b^k_h(s,a) + \mathbb{E}^{s'\sim \hat{P}_{s,a,h}}V^k_{h+1}(s') - r_h(s,a) - \mathbb{E}^{s'\sim P_{s,a,h}}V^*_{h+1}(s')$$
$$=(\hat{r}^k_h(s,a) - r_h(s,a)) + (\mathbb{E}^{s'\sim \hat{P}_{s,a,h}}[V^k_{h+1}(s') - V^*_{h+1}(s')] - \mathbb{E}^{s'\sim P_{s,a,h}}[V^k_{h+1}(s') - V^*_{h+1}(s')])$$
$$+ (\mathbb{E}^{s'\sim \hat{P}_{s,a,h}}V^*_{h+1}(s') - \mathbb{E}^{s'\sim P_{s,a,h}}V^*_{h+1}(s')) + \mathbb{E}^{s'\sim P_{s,a,h}}[V^k_{h+1}(s') - V^*_{h+1}(s')] + b^k_h(s,a).$$

By Lemmas 6 and 7 (with $V = V^k_{h+1} - V^*_{h+1}$) and definition of $b^k_h(s,a)$, we conclude

$$V^k_h(s) - V^*_h(s) \leqslant \mathbb{E}^{s'\sim P_{s,a,h}}[V^k_{h+1}(s') - V^*_{h+1}(s')] + \left(\sqrt{\frac{2\mathbb{V}^{r'\sim R_{s,a,h}}[r']\iota}{n^k_h(s,a)}} + \frac{H\iota}{n^k_h(s,a)}\right)$$

$$+ \left(\sqrt{\frac{2S\mathbb{E}^{s'\sim P_{s,a,h}}[(V^k_{h+1}(s') - V^*_{h+1}(s'))^2]\iota}{n^k_h(s,a)}} + \frac{SH\iota}{n^k_h(s,a)}\right)$$

$$+ \left(\sqrt{\frac{2\mathbb{V}^{s'\sim P_{s,a,h}}[V^*_{h+1}(s')^2]\iota}{n^k_h(s,a)}} + \frac{H\iota}{n^k_h(s,a)}\right)$$

$$+ \left(2\sqrt{\frac{\mathbb{V}^{s'\sim \hat{P}^k_{s,a,h}}[V_{h+1}(s')]\iota}{n^k_h(s,a)}} + 2\sqrt{\frac{(\hat{\sigma}^k_h(s,a) - (\hat{r}^k_h(s,a))^2)\iota}{n^k_h(s,a)}} + \frac{10H\iota}{n^k_h(s,a)}\right)$$

$$\leqslant \mathbb{E}^{s'\sim P_{s,a,h}}[V^k_{h+1}(s') - V^*_{h+1}(s')] + (3\sqrt{2} + 4)H\sqrt{\frac{S\iota}{n^k_h(s,a)}} + \frac{13SH\iota}{n^k_h(s,a)}.$$

If $S\iota \leqslant n_h^k(s, a)$ then we have

$$V_h^k(s) - V_h^*(s) \leqslant \mathbb{E}^{s' \sim P_{s,a,h}}[V_{h+1}^k(s') - V_{h+1}^*(s')] + \left(H \wedge 22H\sqrt{\frac{S\iota}{n_h^k(s, a)}}\right). \quad (15)$$

If $S\iota > n_h^k(s, a)$, then Equation (15) also holds since $V_h^k(s) - V_h^*(s) \leqslant H$. Thus, we can recursively apply Equation (15) and conclude

$$V_h^k(s) - V_h^*(s) \leqslant \mathbb{E}^{\pi^k}\left[\sum_{h'=h}^{H} H \wedge 22H\sqrt{\frac{S\iota}{n_{h'}^k(s_{h'}, a_{h'})}}\,\middle|\, s_h = s\right]$$

$\square$

**Lemma 16.** *Conditioned on success of Lemma 6, if $a = \pi_h^k(s)$ then*

$$E_h^k(s, a) \leqslant \left(H \wedge 5\sqrt{\frac{\mathtt{Var}_h^*(s, a)\iota}{n_h^k(s, a)}}\right) + \mathbb{E}^{\pi^k}\left[\sum_{h'=h}^{H} 3H^2 \wedge \frac{1500SH^2\iota}{n_{h'}^k(s_{h'}, a_{h'})}\,\middle|\, (s_h, a_h) = (s, a)\right].$$

*Proof.* Similar to the proof of Lemma 15,

$$E_h^k(s, a) = Q_h^k(s, a) - r_h(s, a) - \mathbb{E}^{s' \sim P_{s,a,h}}[V_{h+1}^k(s')]$$

$$= \hat{r}_h^k(s, a) + b_h^k(s, a) + \mathbb{E}^{s' \sim \hat{P}_{s,a,h}}V_{h+1}^k(s') - r_h(s, a) - \mathbb{E}^{s' \sim P_{s,a,h}}V_{h+1}^k(s')$$

$$\leqslant |\hat{r}_h^k(s, a) - r_h(s, a)| + |\mathbb{E}^{s' \sim \hat{P}_{s,a,h}}[V_{h+1}^k(s') - V_{h+1}^*(s')] - \mathbb{E}^{s' \sim P_{s,a,h}}[V_{h+1}^k(s') - V_{h+1}^*(s')]|$$

$$+ |\mathbb{E}^{s' \sim \hat{P}_{s,a,h}}V_{h+1}^*(s') - \mathbb{E}^{s' \sim P_{s,a,h}}V_{h+1}^*(s')| + b_h^k(s, a).$$

We will apply Lemmas 6, 7 and 14 to bound each term. So

$$E_h^k(s, a) \leqslant \sqrt{\frac{2\mathbb{V}^{r' \sim R_{s,a,h}}[r']\iota}{n_h^k(s, a)}} + \frac{H\iota}{n_h^k(s, a)}$$

$$+ \frac{1}{H}\mathbb{E}^{s' \sim P_{s,a,h}}[(V_{h+1}^k(s') - V_{h+1}^*(s'))^2] + \frac{SH\iota}{n_h^k(s, a)}$$

$$+ \sqrt{\frac{2\mathbb{V}^{s' \sim P_{s,a,h}}[V_h^*(s')]\iota}{n_h^k(s, a)}} + \frac{H\iota}{n_h^k(s, a)}$$

$$+ \frac{2}{H}\mathbb{E}^{s' \sim \hat{P}_{s,a,h}^k}[(V_{h+1}^k(s') - V_{h+1}^*(s'))^2] + 2\sqrt{\frac{2\mathtt{Var}_h^*(s, a)\iota}{n_h^k(s, a)}} + \frac{20H\iota}{n_h^k(s, a)}.$$

By Lemma 7 (with $V = (V_{h+1}^k - V_{h+1}^*)^2$) again,

$$\mathbb{E}^{s' \sim \hat{P}_{s,a,h}^k}[(V_{h+1}^k(s') - V_{h+1}^*(s'))^2] - \mathbb{E}^{s' \sim P_{s,a,h}}[(V_{h+1}^k(s') - V_{h+1}^*(s'))^2]$$

$$\leqslant \sqrt{\frac{2S\mathbb{E}^{s' \sim P_{s,a,h}}[(V_{h+1}^k(s') - V_{h+1}^*(s'))^4]\iota}{n_h^k(s, a)}} + \frac{SH^2\iota}{n_h^k(s, a)}$$

$$\leqslant \sqrt{\frac{2H^2S\mathbb{E}^{s' \sim P_{s,a,h}}[(V_{h+1}^k(s') - V_{h+1}^*(s'))^2]\iota}{n_h^k(s, a)}} + \frac{SH^2\iota}{n_h^k(s, a)}$$

$$\leqslant \mathbb{E}^{s' \sim P_{s,a,h}}[(V_{h+1}^k(s') - V_{h+1}^*(s'))^2] + \frac{2SH^2\iota}{n_h^k(s, a)}.$$

By Lemma 15,

$$(V_{h+1}^k(s') - V_{h+1}^*(s'))^2 \leqslant \mathbb{E}^{\pi^k}\left[\left(\sum_{h'=h+1}^{H} H \wedge 22H\sqrt{\frac{S\iota}{n_{h'}^k(s_{h'}, a_{h'})}}\right)^2\,\middle|\, s_{h+1} = s'\right]$$

$$\leqslant \mathbb{E}^{\pi^k}\left[\sum_{h'=h+1}^{H} H^3 \wedge \frac{500SH^3\iota}{n_{h'}^k(s_{h'}, a_{h'})}\bigg| s_{h+1} = s'\right].$$

Hence,

$$E_h^k(s, a) \leqslant (2 + 2\sqrt{2})\sqrt{\frac{\mathtt{Var}_h^*(s,a)\iota}{n_h^k(s,a)}} + \frac{3}{H}\mathbb{E}^{s' \sim P_{s,a,h}}[(V_{h+1}^k(s') - V_{h+1}^*(s'))^2] + \frac{27SH\iota}{n_h^k(s,a)}$$

$$\leqslant 5\sqrt{\frac{\mathtt{Var}_h^*(s,a)\iota}{n_h^k(s,a)}} + \mathbb{E}^{\pi^k}\left[\sum_{h'=h}^{H} 3H^2 \wedge \frac{1500SH^2\iota}{n_{h'}^k(s_{h'}, a_{h'})}\bigg|(s_h, a_h) = (s, a)\right].$$

The extra "$H\wedge$" part is because $E_h^k(s, a) \leqslant H$ by definition. $\qquad\square$

### C.3 Concentration of visitation count

This lemma shows that the sum of visiting probabilities is bounded by $n_h^k$.

**Lemma 17.** *With probability at least $1 - \delta$,*

$$\sum_{k'=1}^{k} \mathbb{E}[\mathbf{1}\{(s_h^{k'}, a_h^{k'}) = (s, a)\}|\mathcal{F}_{k'}] \leqslant 3n_h^k(s, a) + \iota$$

*for all $s, a, h, k$, where $\mathcal{F}_k$ is the $\sigma$-field generated by the first $k - 1$ episodes.*

*Proof.* This is a direct consequence of Lemma 3. $\qquad\square$

The next lemma considers a weighted sum over visiting probabilities.

**Lemma 18.** *With probability at least $1 - 2\delta$,*

$$\sum_{k'=1}^{k}\sum_{h'=1}^{h} w_h(s_{h'}^{k'}, a_{h'}^{k'})\mathbb{E}[\mathbf{1}\{(s_h^{k'}, a_h^{k'}) = (s, a)\}|\mathcal{F}_{k',h'}] \leqslant 9\bar{W}n_h^k(s, a) + 4H\bar{W}\iota,$$

*for all $s, a, h, k$, where we recall the definition*

$$w_h(s, a) = \mathtt{Var}_h^*(s, a), \bar{W} = \min\{160H^2\log(4K(H + 1)/\delta), \mathtt{Var}_{\max}^{\mathrm{c}}\}$$

*and $\mathcal{F}_{k,h}$ is the $\sigma$-field generated by the first $(k-1)$ episodes and the first $h$ steps of the $k$-th episode.*

*Proof.* By Lemma 3 and $w_h(s_{h'}^k, a_{h'}^k) \leqslant \bar{W}$,

$$\sum_{k'=1}^{k}\sum_{h'=1}^{h} w_h(s_{h'}^{k'}, a_{h'}^{k'})\mathbb{E}[\mathbf{1}\{(s_h, a_h) = (s, a)\}|\mathcal{F}_{k',h'}] \leqslant 3\sum_{k'=1}^{k}\sum_{h'=1}^{h} w_h(s_{h'}^{k'}, a_{h'}^{k'})\mathbf{1}\{(s_h, a_h) = (s, a)\} + \bar{W}\iota$$

for all $s, a, h, k$. Then, we will bound $\sum_{k'=1}^{k}\sum_{h'=1}^{h} w_h(s_{h'}^{k'}, a_{h'}^{k'})\mathbf{1}\{(s_h, a_h) = (s, a)\}$ in two different ways for each term in $\bar{W}$.

First, we apply Lemma 3 again with respect to only the sum over $k'$ with $(s_h^{k'}, a_h^{k'}) = (s, a)$. This shows

$$\sum_{k'=1}^{k}\sum_{h'=1}^{h} w_h(s_{h'}^{k'}, a_{h'}^{k'})\mathbf{1}\{(s_h^{k'}, a_h^{k'}) = (s, a)\} \leqslant 3\mathtt{Var}_{\max}^{\mathrm{c}}n_h^k(s, a) + H\mathtt{Var}_{\max}^{\mathrm{c}}\iota.$$

Second, by Lemma 8, with probability $1 - \delta$,

$$\sum_{h'=1}^{H} w_h(s_{h'}^{k'}, a_{h'}^{k'})\mathbf{1}\{(s_h^{k'}, a_h^{k'}) = (s, a)\} \leqslant 160H^2\log(4K(H + 1)/\delta)\mathbf{1}\{(s_h^{k'}, a_h^{k'}) = (s, a)\}$$

for all $k' = 1, 2, \cdots, K$. Thus,

$$\sum_{k'=1}^{k} \sum_{h'=1}^{h} w_h(s_{h'}^{k'}, a_{h'}^{k'}) \mathbf{1}\{(s_h^{k'}, a_h^{k'}) = (s, a)\} \leqslant 160 H^2 \log(4K(H+1)/\delta) n_h^k(s, a).$$

Hence we conclude

$$\sum_{k'=1}^{k} \sum_{h'=1}^{h} w_h(s_{h'}^{k'}, a_{h'}^{k'}) \mathbb{E}[\mathbf{1}\{(s_h^{k'}, a_h^{k'}) = (s, a)\} | \mathcal{F}_{k', h'}] \leqslant 9\bar{W} n_h^k(s, a) + 4H\bar{W}\iota.$$

$\square$

**Lemma 19.** *Let $f$ be a non-increasing nonnegative function defined on $[1, +\infty)$ with $f(1) \leqslant M$. Conditioned on success event of Lemma 17, we have*

$$\sum_{k=1}^{K} f(n_h^k(s, a)) \mathbb{P}[(s_h^k, a_h^k) = (s, a) | \mathcal{F}_k] \leqslant M(\iota + 3) + 3 \int_1^{n_h^K(s,a)} f(x) \mathrm{d}x.$$

*Proof.* Let $n_k' = \sum_{k'=1}^{k} \mathbb{P}[(s_h^{k'}, a_h^{k'}) = (s, a) | \mathcal{F}_{k'}] \leqslant 3 n_h^k(s, a) + \iota$ and

$$K_0 = \min\{k : n_k' \geqslant \iota + 3\}.$$

(If $n_K' < \iota + 3$ then we define $K_0 = K$.) Then,

$$\sum_{k=1}^{K} f(n_h^k(s, a)) \mathbb{P}[(s_h^k, a_h^k) = (s, a) | \mathcal{F}_k]$$

$$= \sum_{k=1}^{K_0} f(n_h^k(s, a)) \mathbb{P}[(s_h^k, a_h^k) = (s, a) | \mathcal{F}_k] + \sum_{k=K_0+1}^{K} f(n_h^k(s, a)) \mathbb{P}[(s_h^k, a_h^k) = (s, a) | \mathcal{F}_k]$$

$$\leqslant M \sum_{k=1}^{K_0} \mathbb{P}[(s_h^k, a_h^k) = (s, a) | \mathcal{F}_k] + \sum_{k=K_0+1}^{K} f((n_k' - \iota)/3)(n_k' - n_{k-1}')$$

$$\leqslant M n_{K_0}' + \sum_{k=K_0+1}^{K} \int_{n_{k-1}'}^{n_k'} f((x - \iota)/3) \mathrm{d}x$$

$$\leqslant M(\iota + 3) + \int_{n_{K_0}'}^{n_K'} f((x - \iota)/3) \mathrm{d}x \leqslant M(\iota + 3) + \int_{\iota+3}^{n_K'} f((x - \iota)/3) \mathrm{d}x$$

$$= M(\iota + 3) + 3 \int_1^{(n_K' - \iota)/3} f(x) \mathrm{d}x \leqslant M(\iota + 3) + 3 \int_1^{n_h^K(s,a)} f(x) \mathrm{d}x.$$

$\square$

**Lemma 20.** *Let $f$ be a non-increasing nonnegative function defined on $[0, +\infty)$ with upper bound $M$. Conditioned on success event of Lemma 18, we have*

$$\sum_{k=1}^{K} f(n_h^k(s, a)) \sum_{h'=1}^{h} w_h(s_{h'}^k, a_{h'}^k) \mathbb{P}[(s_h^k, a_h^k) = (s, a) | \mathcal{F}_{k, h'}] \leqslant M\bar{W}(4H\iota + 9) + 9\bar{W} \int_1^{n_h^K(s,a)} f(x) \mathrm{d}x.$$

The proof is similar to that of Lemma 19.

## C.4 Final calculations

Our calculations are conditioned on success of Lemmas 6, 9, 17 and 18, and they happen simultaneously with probability at least $1 - 20\delta$.

We begin by analyzing the clipped surplus. By Lemmas 5 and 16, we have

$$\bar{E}_h^k(s, a)$$

$$\leqslant 2\,\mathrm{clip}\left[H \wedge 5\sqrt{\frac{\mathtt{Var}_h^*(s,a)\iota}{n_h^k(s,a)}} \Big| \frac{\Delta_{\min}\mathtt{Var}_h^*(s,a)}{24(H^2 \wedge \mathtt{Var}_{\max}^{\mathrm{c}})}\right]$$

$$+ 2\,\mathrm{clip}\left[\mathbb{E}^{\pi^k}\left[\sum_{h'=h}^{H} 3H^2 \wedge \frac{1500SH^2\iota}{n_{h'}^k(s_{h'},a_{h'})}\Big|(s_h,a_h) = (s,a)\right] \Big| \frac{\Delta_{\min}}{24H^2}\right]$$

$$= 2\sum_{s',a'} \mathbf{1}\{(s',a') = (s,a)\}\,\mathrm{clip}\left[H \wedge 5\sqrt{\frac{\mathtt{Var}_h^*(s',a')\iota}{n_h^k(s',a')}} \Big| \frac{\Delta_{\min}\mathtt{Var}_h^*(s',a')}{24(H^2 \wedge \mathtt{Var}_{\max}^{\mathrm{c}})}\right]$$

$$+ 2\,\mathrm{clip}\left[\sum_{s',a'}\sum_{h'=h}^{H}\mathbb{E}^{\pi^k}\left[\left(3H^2 \wedge \frac{1500SH^2\iota}{n_{h'}^k(s',a')}\right)\mathbf{1}\{(s_{h'},a_{h'}) = (s',a')\}\Big|(s_h,a_h) = (s,a)\right] \Big| \frac{\Delta_{\min}}{24H^2}\right]$$

$$\leqslant 2\sum_{s',a'} \mathbf{1}\{(s',a') = (s,a)\}f_h(s,a;n_h^k(s,a))$$

$$+ 4\sum_{s',a'}\sum_{h'=h}^{H}\mathrm{clip}\left[\left(3H^2 \wedge \frac{1500SH^2\iota}{n_{h'}^k(s,a)}\right)\mathbb{E}^{\pi^k}\left[\mathbf{1}\{(s_{h'},a_{h'}) = (s,a)\}|(s_h,a_h) = (s,a)\right] \Big| \frac{\Delta_{\min}}{48SAH^3}\right]$$

$$\leqslant 2\sum_{s',a'} \mathbf{1}\{(s',a') = (s,a)\}f_h(s,a;n_h^k(s,a))$$

$$+ 4\sum_{'s,a'}\sum_{h'=h}^{H}\mathbb{E}^{\pi^k}\left[\mathbf{1}\{(s,a) = (s_{h'},a_{h'})\}|(s_h,a_h) = (s,a)\right]\mathrm{clip}\left[3H^2 \wedge \frac{1500SH^2\iota}{n_{h'}^k(s,a)}\Big| \frac{\Delta_{\min}}{48SAH^3}\right]$$

$$= 2\sum_{s',a'} \mathbf{1}\{(s',a') = (s,a)\}f_h(s,a;n_h^k(s,a))$$

$$+ 4\sum_{s',a'}\sum_{h'=h}^{H}\mathbb{P}^{\pi^k}\left[(s',a') = (s_{h'},a_{h'})|(s_h,a_h) = (s,a)\right]g(n_{h'}^k(s',a')),$$

where

$$f_h(s,a;x) = \mathrm{clip}\left[H \wedge 5\sqrt{\frac{\mathtt{Var}_h^*(s,a)\iota}{x}} \Big| \frac{\Delta_{\min}\mathtt{Var}_h^*(s,a)}{24(H^2 \wedge \mathtt{Var}_{\max}^{\mathrm{c}})}\right], \quad g(x) = \mathrm{clip}\left[3H^2 \wedge \frac{1500SH^2\iota}{x}\Big| \frac{\Delta_{\min}}{48SAH^3}\right].$$

### C.4.1 Bounding regret

By Lemma 13, we have

$$\mathrm{Regret}(K) \leqslant \frac{3}{2}\sum_{k=1}^{K}\sum_{h=1}^{H}\mathbb{E}^{\pi^k}\left[\bar{E}_h^k(s_h,a_h)\right] = \frac{3}{2}\sum_{k=1}^{K}\sum_{h=1}^{H}\mathbb{E}[\bar{E}_h^k(s_h^k,a_h^k)|\mathcal{F}_k]$$

$$\leqslant 3\sum_{k=1}^{K}\sum_{h=1}^{K}\sum_{s,a}\mathbb{E}\left[\mathbf{1}\{(s,a) = (s_h^k,a_h^k)\}f_h(s,a;n_h^k(s,a))|\mathcal{F}_k\right]$$

$$+ 6\sum_{k=1}^{K}\sum_{h=1}^{K}\sum_{s,a}\sum_{h'=h}^{H}\mathbb{E}\left[\mathbb{P}^{\pi^k}\left[(s,a) = (s_{h'},a_{h'})|(s_h,a_h) = (s_h^k,a_h^k)\right]g(n_{h'}^k(s,a))|\mathcal{F}_k\right]$$

$$= 3\sum_{s,a}\sum_{h=1}^{K}\sum_{k=1}^{K}f_h(s,a;n_h^k(s,a))\mathbb{P}[(s,a) = (s_h^k,a_h^k)|\mathcal{F}_k]$$

$$+ 6\sum_{h=1}^{H}\sum_{s,a}\sum_{h'=h}^{H}\sum_{k=1}^{K}g(n_{h'}^k(s,a))\mathbb{P}\left[(s,a) = (s_{h'}^k,a_{h'}^k)|\mathcal{F}_k\right].$$

We will use Lemma 19 to bound the two sums.

For the first sum, we have then

$$\sum_{k=1}^{K}f_h(s,a;n_h^k(s,a))\mathbb{P}[(s,a) = (s_h^k,a_h^k)] \leqslant H(\iota+3) + 3\int_{1}^{n_h^K(s,a)}f_h(s,a;n_h^k(s,a))\mathrm{d}x.$$

When $\Delta_h(s,a) > 0$, we bound the integration by

$$\int_1^{n_h^K(s,a)} f_h(s,a;n_h^k(s,a))\mathrm{d}x = \int_1^{n_h^K(s,a)} \mathrm{clip}\left[5\sqrt{\frac{\mathtt{Var}_h^*(s,a)\iota}{x}}|\frac{\Delta_{\min}\mathtt{Var}_h^*(s,a)}{24(H^2 \wedge \mathtt{Var}_{\max}^{\mathrm{c}})}\right]\mathrm{d}x$$

$$\leqslant \int_0^{n_h^K(s,a)} 5\sqrt{\frac{\mathtt{Var}_h^*(s,a)\iota}{x}}\mathrm{d}x = 10\sqrt{\mathtt{Var}_h^*(s,a)n_h^K(s,a)\iota}.$$

When $\Delta_h(s,a) = 0$, we bound by

$$\int_1^{n_h^K(s,a)} f_h(s,a;n_h^k(s,a))\mathrm{d}x = \int_1^{n_h^K(s,a)} \mathrm{clip}\left[5\sqrt{\frac{\mathtt{Var}_h^*(s,a)\iota}{x}}|\frac{\Delta_{\min}\mathtt{Var}_h^*(s,a)}{24(H^2 \wedge \mathtt{Var}_{\max}^{\mathrm{c}})}\right]\mathrm{d}x$$

$$\leqslant \int_0^{+\infty} \mathrm{clip}\left[5\sqrt{\frac{\mathtt{Var}_h^*(s,a)\iota}{x}}|\frac{\Delta_{\min}\mathtt{Var}_h^*(s,a)}{24(H^2 \wedge \mathtt{Var}_{\max}^{\mathrm{c}})}\right]\mathrm{d}x$$

$$= \int_0^{(120(H^2 \wedge \mathtt{Var}_{\max}^{\mathrm{c}}))^2\iota/\Delta_{\min}^2\mathtt{Var}_h^*(s,a)} 5\sqrt{\frac{\mathtt{Var}_h^*(s,a)\iota}{x}}\mathrm{d}x$$

$$= \frac{1200(H^2 \wedge \mathtt{Var}_{\max}^{\mathrm{c}})\iota}{\Delta_{\min}}.$$

For the second sum, we bound similarly that

$$\sum_{h'=h}^H g(n_{h'}^k(s,a))\mathbb{P}\left[(s,a) = (s_{h'}^k, a_{h'}^k)\right]$$

$$\leqslant 3H^2(\iota + 3) + 3\int_1^{n_h^K(s,a)} \mathrm{clip}\left[\frac{1500SH^2\iota}{x}|\frac{\Delta_{\min}}{48SAH^3}\right]\mathrm{d}x$$

$$\leqslant 3H^2(\iota + 3) + 3\int_1^{72000S^2AH^5\iota/\Delta_{\min}} \frac{1500SH^2\iota}{x}\mathrm{d}x$$

$$= 3H^2(\iota + 3) + 4500SH^2\iota \log(72000S^2AH^5\iota/\Delta_{\min}).$$

Thus,

$$\mathrm{Regret}(K) \leqslant 3\sum_{(s,a,h)\in\mathcal{Z}_{\mathrm{sub}}} (4H\iota + 30\sqrt{\mathtt{Var}_h^*(s,a)n_h^K(s,a)\iota}) + 3\sum_{(s,a,h)\in\mathcal{Z}_{\mathrm{opt}}} \left(4H\iota + \frac{3600(H^2 \wedge \mathtt{Var}_{\max}^{\mathrm{c}})\iota}{\Delta_{\min}}\right)$$

$$+ 6SAH^2(12H^2 + 4500SH^2\iota \log(72000S^2AH^5\iota/\Delta_{\min}))$$

$$\leqslant 90\sum_{(s,a,h)\in\mathcal{Z}_{\mathrm{sub}}} \sqrt{\mathtt{Var}_h^*(s,a)n_h^K(s,a)\iota} + 10800\sum_{(s,a,h)\in\mathcal{Z}_{\mathrm{opt}}} \frac{(H^2 \wedge \mathtt{Var}_{\max}^{\mathrm{c}})\iota}{\Delta_{\min}}$$

$$+ 27000S^2AH^4\iota \log(72000S^2AH^5\iota/\Delta_{\min}) + 96SAH^5\iota. \tag{16}$$

### C.4.2  Lower-bounding visitation count

Recall the lower bound in Lemma 12. We begin by taking a weighted sum over all states visited during the algorithm:

$$\sum_{k=1}^K \sum_{h=1}^H w_h(s_h^k, a_h^k)\Delta_h(s_h^k, a_h^k) \leqslant \frac{3}{2}\sum_{k=1}^K \sum_{h=1}^H w_h(s_h^k, a_h^k) \sum_{h'=h}^H \mathbb{E}[\bar{E}_{h'}^k(s_{h'}^k, a_{h'}^k)|\mathcal{F}_{k,h}]$$

$$\leqslant 3\sum_{k=1}^K \sum_{h=1}^H w_h(s_h^k, a_h^k) \sum_{h'=h}^H \sum_{s,a} \mathbb{E}\left[\mathbf{1}\{(s,a) = (s_{h'}^k, a_{h'}^k)\}f_h(s,a;n_{h'}^k(s,a))|\mathcal{F}_{k,h}\right]$$

$$+ 6\sum_{k=1}^K \sum_{h=1}^K w_h(s_h^k, a_h^k) \sum_{h'=h}^H \sum_{s,a} \sum_{h*=h'}^H \mathbb{E}\left[\mathbb{P}^{\pi^k}\left[(s,a) = (s_{h*}, a_{h*})|(s_{h'}, a_{h'}) = (s_{h'}^k, a_{h'}^k)\right] g(n_{h*}^k(s,a))|\mathcal{F}_{k,h}\right]$$

$$=3\sum_{s,a}\sum_{h'=1}^{H}\sum_{k=1}^{K}f_h(s,a;n_{h'}^k(s,a))\sum_{h=1}^{h'}w_h(s_h^k,a_h^k)\mathbb{P}\left[(s,a)=(s_{h'}^k,a_{h'}^k)|\mathcal{F}_{k,h}\right]$$

$$+6\sum_{s,a}\sum_{h*=1}^{H}\sum_{k=1}^{K}g(n_{h*}^k(s,a))\sum_{h=1}^{h*}\sum_{h'=h}^{h*}w_h(s_h^k,a_h^k)\mathbb{P}[(s,a)=(s_{h*}^k,a_{h*}^k)|\mathcal{F}_{k,h}]$$

$$=3\sum_{s,a}\sum_{h'=1}^{H}\sum_{k=1}^{K}f_h(s,a;n_{h'}^k(s,a))\sum_{h=1}^{h'}w_h(s_h^k,a_h^k)\mathbb{P}\left[(s,a)=(s_{h'}^k,a_{h'}^k)|\mathcal{F}_{k,h}\right]$$

$$+6H\sum_{s,a}\sum_{h*=1}^{H}\sum_{k=1}^{K}g(n_{h*}^k(s,a))\sum_{h=1}^{h*}w_h(s_h^k,a_h^k)\mathbb{P}[(s,a)=(s_{h*}^k,a_{h*}^k)|\mathcal{F}_{k,h}].$$

Take $w_h(s,a)=\mathtt{Var}_h^*(s,a)$, it follows from Lemma 20 that

$$\sum_{s,a}\sum_{h=1}^{H}w_h(s,a)\Delta_h(s,a)n_h^k(s,a)=\sum_{k=1}^{K}\sum_{h=1}^{H}w_h(s_h^k,a_h^k)\Delta_h(s_h^k,a_h^k)$$

$$\leqslant 3\sum_{s,a}\sum_{h'=1}^{H}\bar{W}\left(3H^2(4H\iota+9)+9\int_1^{n_h^K(s,a)}f_{h'}(s,a;x)\mathrm{d}x\right)$$

$$+6H\sum_{s,a}\sum_{h*=1}^{H}\bar{W}\left(3H^2(4H\iota+9)+9\int_1^{n_h^K(s,a)}f_{h*}(s,a;x)\mathrm{d}x\right)$$

$$\leqslant 351\bar{W}SAH^5\iota+270\bar{W}\sum_{(s,a,h)\in\mathcal{Z}_{\mathrm{sub}}}\sqrt{\mathtt{Var}_h^*(s,a)n_h^K(s,a)\iota}$$

$$+\frac{32400|\mathcal{Z}_{\mathrm{opt}}|\bar{W}(H^2\wedge\mathtt{Var}_{\mathrm{max}}^{\mathrm{c}})\iota}{\Delta_{\mathrm{min}}}+81000S^2AH^4\iota\bar{W}\log(72000S^2AH^5\iota/\Delta_{\mathrm{min}}).\quad(17)$$

For notational simplicity, we denote

$$R_0=\sum_{(s,a,h)\in\mathcal{Z}_{\mathrm{sub}}}\sqrt{\mathtt{Var}_h^*(s,a)\Delta_h(s,a)\iota}.$$

By Equation (17) and Cauchy-Schwarz inequality,

$$\bar{W}\left(351SAH^5\iota+270R_0+\frac{32400|\mathcal{Z}_{\mathrm{opt}}|(H^2\wedge\mathtt{Var}_{\mathrm{max}}^{\mathrm{c}})\iota}{\Delta_{\mathrm{min}}}+81000S^2AH^4\iota\log(72000S^2AH^5\iota/\Delta_{\mathrm{min}})\right)$$

$$\cdot\left(\sum_{(s,a,h)\in\mathcal{Z}_{\mathrm{sub}}}\frac{\iota}{\Delta_h(s,a)}\right)\geqslant R_0^2.$$

It follows by solving the quadratic equation that

$$R_0\leqslant\sum_{(s,a,h)\in\mathcal{Z}_{\mathrm{sub}}}\frac{540\bar{W}\iota}{\Delta_h(s,a)}+2SAH^5\iota+\frac{120|\mathcal{Z}_{\mathrm{opt}}|(H^2\wedge\mathtt{Var}_{\mathrm{max}}^{\mathrm{c}})\iota}{\Delta_{\mathrm{min}}}+300S^2AH^4\iota\log(72000S^2AH^5\iota/\Delta_{\mathrm{min}}).$$

From Equation (16),

$$\mathrm{Regret}(K)\leqslant 90R_0+96SAH^4\iota+10800\sum_{(s,a,h)\in Zopt}\frac{(H^2\wedge\mathtt{Var}_{\mathrm{max}}^{\mathrm{c}})\iota}{\Delta_{\mathrm{min}}}+27000S^2AH^3\iota\log(72000S^2AH^5\iota/\Delta_{\mathrm{min}})$$

$$\leqslant\sum_{(s,a,h)\in\mathcal{Z}_{\mathrm{sub}}}\frac{48600\bar{W}\iota}{\Delta_h(s,a)}+\frac{21600|\mathcal{Z}_{\mathrm{opt}}|(H^2\wedge\mathtt{Var}_{\mathrm{max}}^{\mathrm{c}})\iota}{\Delta_{\mathrm{min}}}+270000S^2AH^4\iota\log(10SAH\iota/\Delta_{\mathrm{min}})+276SAH^5\iota,$$

with probability at least $1-20\delta$, as we have claimed in the main text.

Thus we have proved the following main theorem.

**Theorem 5** (Formal statement of Theorem 2). *Suppose we run MVP algorithm with universal constants $c_1 = c_2 = 2, c_3 = 10$. For any MDP instance $\mathcal{M}$ satisfying Assumption 1 and any confidence parameter $\delta > 0$, any episode number $K \geqslant 1$, with probability at least $1 - 20\delta$,*

$$\mathrm{Regret}(K) \lesssim \sum_{(s,a,h)\in\mathcal{Z}_{\mathrm{sub}}} \frac{(H^2 \log(HK/\delta) \wedge \mathtt{Var}_{\max}^{\mathrm{c}}) \log(SAHK/\delta)}{\Delta_h(s,a)}$$

$$+ \frac{|\mathcal{Z}_{\mathrm{opt}}|(H^2 \wedge \mathtt{Var}_{\max}^{\mathrm{c}}) \log(SAHK/\delta)}{\Delta_{\min}}$$

$$+ S^2 AH^4 \log(SAHK/\delta) \log(SAH\Delta_{\min}^{-1}\log(SAHK/\delta))$$

$$+ SAH^5 \log(SAHK/\delta).$$

# D  Regret Lower Bound

**Theorem 6** (Formal statement of Theorem 3). *For a given configuration of $S, A, H$, target conditional variance $L \in [1, H^2]$, as well as a set of suboptimality gaps $\boldsymbol{\Delta} = \{\Delta_1, \Delta_2, \ldots, \Delta_{SAH}\}$, we make the following mild assumptions:*

- *Let $\mathcal{I} = \{i \mid \Delta_i = 0\}$. Assume that $|\mathcal{I}| \geqslant SH$, i.e., the suboptimality gaps are realizable.*

- *Assume that $\Delta_i < \sqrt{L}$ for all $1 \leqslant i \leqslant SAH$.*

*For any algorithm $\boldsymbol{\pi}$, there exists an MDP instance $\mathcal{M}^{\boldsymbol{\pi}}$ satisfying:*

- *It has $|\bar{\mathcal{S}}| = S + 2$ states and $A$ actions.*

- *There exists $\mathcal{S} \subset \bar{\mathcal{S}}$ such that $|\mathcal{S}| = S$, and a bijection $\sigma$ between $[H] \times \mathcal{S} \times \mathcal{A}$ and $[SAH]$, satisfying $\Delta_h(s,a) = \frac{1}{4}\Delta_{\sigma(h,s,a)}$ for any $(h,s,a) \in [H] \times \mathcal{S} \times \mathcal{A}$.*

- $\mathtt{Var}_{\max}^{\mathrm{c}} = \Theta(L)$, *while* $\mathtt{Var}_{\max} \leqslant O(1)$.

*such that*

$$\lim_{K\to\infty} \frac{\mathbb{E}^{\boldsymbol{\pi}}[\mathrm{Regret}(\mathcal{M}^{\boldsymbol{\pi}}, K)]}{\log K} \geqslant \Omega\left(\sum_{i:\Delta_i>0} \frac{L}{\Delta_i}\right).$$

*Proof.* First consider multi-armed bandit lower bound given a set of gaps $\boldsymbol{\Delta} = \{\Delta_1, \Delta_2, \ldots, \Delta_A\}$ and a target variance $L$. WLOG, assume $\Delta_i \leqslant \Delta_{i+1}$. Construct Bernoulli outcomes for each action $a_i$: w.p. $p_i = \frac{1}{2} - \frac{\Delta(a_i)}{4\sqrt{L}} \in [\frac{1}{4}, \frac{1}{2}]$, get reward $\sqrt{L}$; w.p. $1 - p_i$, get reward 0. Then $Q(a_i) = p_i = \frac{1}{2} - \frac{\Delta(a_i)}{4\sqrt{L}}$, and $Q(a_1) - Q(a_i) = \frac{\Delta(a_i)}{4\sqrt{L}}$. Then $\mathtt{Var}(a_i) = p_i(1-p_i)L = \Theta(L)$. We invoke standard lower bound [Lai and Robbins, 1985] with reward outcomes in $[0, 1]$. We first scale the rewards in our example by $\frac{1}{\sqrt{L}}$. For any algorithm $\boldsymbol{\pi}$, there exists a permutation on the gaps (into $\frac{1}{\sqrt{L}}\boldsymbol{\Delta}^{\boldsymbol{\pi}}$), such that

$$\lim_{K\to\infty} \frac{\mathbb{E}[\mathrm{Regret}(\frac{1}{\sqrt{L}}\boldsymbol{\Delta}^{\boldsymbol{\pi}}, K)]}{\log K} \geqslant \sum_i \frac{Q(a_1) - Q(a_i)}{\mathsf{kl}(p_i, \frac{1}{2})} \overset{\text{(i)}}{\geqslant} \sum_{i:\Delta_i>0} \frac{\mathtt{Var}(a_i)}{Q(a_1) - Q(a_i)} \geqslant \Omega\left(\sum_{i:\Delta_i>0} \frac{1}{\Delta_i/\sqrt{L}}\right),$$

where $\mathsf{kl}(p, q) = p\log\frac{p}{q} + (1-p)\log\frac{1-p}{1-q}$; (i) is by $\frac{(\frac{1}{2}-x)^2}{x(1-x)} \geqslant x\log(2x) + (1-x)\log(2-2x)$ for $x \in [0, 1]$ (we take $x = p_i$). To see this, we substitute $t = 1 - 2x \in [-1, 1]$, then

$$\frac{(\frac{1}{2} - x)^2}{x(1-x)} = \frac{t^2}{(1-t)(1+t)} \geqslant t^2$$

and

$$x\log(2x) + (1-x)\log(2-2x) = \frac{1-t}{2}\log(1-t) + \frac{1+t}{2}\log(1+t) \leqslant \frac{-t(1-t)}{2} + \frac{t(1+t)}{2} = t^2.$$

Scaling back, we have

$$\lim_{K\to\infty}\frac{\mathbb{E}[\text{Regret}(\boldsymbol{\Delta}^{\boldsymbol{\pi}},K)]}{\log K}=\lim_{K\to\infty}\frac{\sqrt{L}\mathbb{E}[\text{Regret}(\frac{1}{\sqrt{L}}\boldsymbol{\Delta}^{\boldsymbol{\pi}},K)]}{\log K}\geqslant\Omega\left(\sum_{i:\Delta_i>0}\frac{L}{\Delta(a_i)}\right).$$

Then, we construct the MDP as:

- **States:** in total $S+2$ states. $s_0$ as a main state, $s_1,s_2,\ldots,s_S$ as bandit states, $s_{-1}$ as a terminal state.

- **Transition:** $s_0$ does not require decision-making: $P_{s_0,a,h}(s_0)=1-\frac{1}{LH}$, $P_{s_0,a,h}(s_i)=\frac{1}{LSH}$ for $1\leqslant i\leqslant S$. $s_i$ is a bandit problem, and directly transits into $s_{-1}$: $P_{s_i,a,h}(s_{-1})=1$ for $1\leqslant i\leqslant S$. $s_{-1}$ is self-absorbing: $P_{s_{-1},a,h}(s_{-1})=1$.

- **Rewards:** for $s_0$ and $s_{-1}$, all rewards are 0. Rewards for $(s_i,a,h)$ are decided by the construction below.

Assign $\boldsymbol{\Delta}$ into $H\times S$ groups, each with exactly $A$ items: $\{\boldsymbol{\Delta}_{h,s_i}\}_{(h,i)\in[H]\times[S]}$ and from the assumption we can guarantee at least one 0 gap in each group. We have $d_h(s_i)=\frac{1}{LSH}(1-\frac{1}{LH})^{h-1}\in\left[\frac{1}{\text{e}LSH},\frac{1}{LSH}\right]$ for $1\leqslant i\leqslant S$. For each $(h,i)\times[H]\times[S]$, from Lemma 1, with probability at least $1-\frac{1}{2HS}$,

$$\left|d_h(s_i)-\frac{N_h^K(s_i)}{K}\right|\leqslant\sqrt{\frac{2d_h(s_i)(1-d_h(s_i))\log(4SH)}{K}}+\frac{\log(4SH)}{K}$$

$$\Rightarrow Kd_h(s_i)-N_h^K(s_i)\leqslant\sqrt{\frac{2K}{LSH}\log(4SH)}+\log(4SH).$$

When $K\geqslant 2\text{e}^2(1+\sqrt{1+\text{e}})^2LSH\log(4SH)$, we have RHS $\leqslant\frac{K}{2\text{e}LSH}$, so we have $N_h^k(s_i)\geqslant Kd_h(s_i)-\frac{K}{2\text{e}LSH}\geqslant\frac{K}{2\text{e}LSH}$. Denote the event $\mathcal{E}=\{N_h^K(s_i)\geqslant\frac{K}{2\text{e}LSH}\mid(h,i)\in[H]\times[S]\}$, then $\mathbb{P}[\mathcal{E}]\geqslant\frac{1}{2}$.

Since we set independent random instances for each $(h,s_i)$, we have that

$$\begin{aligned}\lim_{K\to\infty}\frac{\mathbb{E}[\text{Regret}(\boldsymbol{\Delta}_{h,s_i}^{\boldsymbol{\pi}},\boldsymbol{\pi},K)]}{\log K}&\geqslant\lim_{K\to\infty}\frac{\mathbb{P}[\mathcal{E}]\mathbb{E}[\text{Regret}(\boldsymbol{\Delta}_{h,s_i}^{\boldsymbol{\pi}},\frac{K}{2\text{e}LSH})]+(1-\mathbb{P}[\mathcal{E}])\cdot 0}{\log K}\\&\overset{(i)}{\geqslant}\lim_{K\to\infty}\frac{\mathbb{E}[\text{Regret}(\boldsymbol{\Delta}_{h,s_i}^{\boldsymbol{\pi}},\frac{K}{2\text{e}LSH})]}{4\log(\frac{K}{2\text{e}LSH})}\\&\geqslant\Omega\left(\sum_{a:\Delta_{h,s_i}(a)>0}\frac{L}{\Delta_{h,s_i}(a)}\right),\end{aligned}$$

where (i) is by $\mathbb{P}[\mathcal{E}]\geqslant\frac{1}{2}$ and taking $K\geqslant(2\text{e}LSH)^2$. So

$$\begin{aligned}\lim_{K\to\infty}\frac{\mathbb{E}[\text{Regret}(\mathcal{M}^{\boldsymbol{\pi}},\boldsymbol{\pi},K)]}{\log K}&=\lim_{K\to\infty}\sum_{h,i}\frac{\mathbb{E}[\text{Regret}(\boldsymbol{\Delta}_{h,s_i}^{\boldsymbol{\pi}},\boldsymbol{\pi},K)]}{\log K}\\&=\sum_{h,i}\lim_{K\to\infty}\frac{\mathbb{E}[\text{Regret}(\boldsymbol{\Delta}_{h,s_i}^{\boldsymbol{\pi}},\boldsymbol{\pi},K)]}{\log K}\\&\geqslant\Omega\left(\sum_{(h,i,a):\Delta_{h,s_i}(a)>0}\frac{L}{\Delta_{h,s_i}(a)}\right)\\&=\Omega\left(\sum_{i:\Delta_i>0}\frac{L}{\Delta_i}\right).\end{aligned}$$

We have $\text{Var}_h^*(s_0) = \Theta\left((1 - \frac{1}{LH})\frac{1}{LH} \cdot L\right) = \Theta(\frac{1}{H})$, $\text{Var}_h^*(s_{-1}) = 0$, and $\text{Var}_h^*(s_i) = \Theta(L)$. It is easy to verify that $\text{Var}_{\max}^c$ is taken at states $(h, s_i)$, so

$$\text{Var}_{\max}^c = \max_{h,i} \left\{ \text{Var}_h^*(s_i) + \sum_{t=1}^{h-1} \text{Var}_t^*(s_0) \right\} = \Theta(L).$$

However,

$$\text{Var}_{\max} \leqslant \sum_{h=1}^{H} \left( d_h(s_0)\text{Var}_h^*(s_0) + \sum_{i=1}^{S} d_h(s_i)\text{Var}_h^*(s_i) \right) \leqslant O(1),$$

showcasing the separation between $\text{Var}_{\max}^c$ and $\text{Var}_{\max}$. $\qquad\square$

