# OpenReview forum: "Sharp Gap-Dependent Variance-Aware Regret Bounds for Tabular MDPs"
_NeurIPS.cc/2025/Conference — NeurIPS 2025 poster_

### Official Review · Reviewer_hoFY · 2025-06-02

**Clarity:** 4
**Significance:** 4
**Originality:** 4
**Rating:** 5
**Confidence:** 4

**Summary:**

This paper investigates gap-dependent regret bounds for tabular episodic Markov Decision Processes (MDPs).

The authors demonstrate that the previously proposed MVP algorithm can attain a variance-aware, gap-dependent regret bound.

They also establish a matching lower bound, highlighting the tightness and complementarity of both the upper and lower bounds.

The proofs use a novel analytical approach.

**Questions:**

In the abstract, should the definition of delta_min be restricted to (h,s,a) that delta_h(s,a) > 0 as is discussed in more detail later in the part of the paper?

In Definition 2, the summation starts from h'=1 and it is conditioned on a potentially later step s_h, which feels somewhat unconventional. Could you provide some intuition on why this formulation is important and how it plays a role in the regret bound?

In line 181, should the reference be to Definition 2?

I’m particularly interested in the gap between the upper and lower bounds presented in the paper. Line 261 refers to Appendix H for further discussion, but I couldn’t locate it in the supplementary material. Could you provide clarification or additional insights on this point?

**Ethical Concerns:**

["NO or VERY MINOR ethics concerns only"]

**Final Justification:**

It seems that there is no major questions regarding the work, and this work makes a great contribution.

I find the rebuttal satisfactory and will keep my score.

**Limitations:**

not needed

**Paper Formatting Concerns:**

not needed

**Quality:**

4

**Strengths And Weaknesses:**

This paper effectively identifies an important gap in the RL literature: achieving the tightest problem-dependent regret bounds while accounting for both variance and gap.

The concept of maximum conditional total variance is novel to me.

The writing is clear and accessible, making the main contributions easy to understand. The related work section provides a well-structured comparison with prior approaches, helping readers contextualize the significance of the results.

New notations and concepts are introduced in a clear and organized manner. The overall structure of the paper (such as including a proof highlights section in the main text, as well as dedicated sections for notation and technical lemmas in the appendix) greatly enhances readability.

One limitation is that the MVP algorithm is restricted to tabular MDPs. It would be exciting to see future work that extends these ideas to more general settings beyond the tabular case.

---

> ### Author Rebuttal · Authors · 2025-07-31
>
> > In the abstract, should the definition of delta_min be restricted to (h,s,a) that delta_h(s,a) > 0 as is discussed in more detail later in the part of the paper?
>
> > In line 181, should the reference be to Definition 2?
>
> > I’m particularly interested in the gap between the upper and lower bounds presented in the paper. Line 261 refers to Appendix H for further discussion, but I couldn’t locate it in the supplementary material. Could you provide clarification or additional insights on this point?
>
> We thank the reviewer for identifying these typographical and referencing issues. We will correct them in the camera-ready version, should the paper be accepted. Specifically:
>
> - In the abstract, the definition of $\Delta_{\min}$ is indeed intended to apply only to $(s,a,h)$ tuples with $\Delta_h(s,a) > 0$, consistent with the more detailed discussion later in the paper.
>
> - The citation on line 181 should refer to Definition 2, not Definition 1.
>
> - The reference on line 261 should point to Appendix C, not Appendix H. Appendix C contains the formal proof of Theorem 3 and elaborates on the gap between the upper and lower bounds.
>
> > In Definition 2, the summation starts from h'=1 and it is conditioned on a potentially later step s_h, which feels somewhat unconventional. Could you provide some intuition on why this formulation is important and how it plays a role in the regret bound?
>
> We agree that the formulation of conditional total variance in Definition 2 may appear unconventional, as it conditions on a future state $s_h$, which breaks the usual Markov structure. However, this definition is crucial to our analysis.
>
> The summation from $h'=1$ to $h$ quantifies the variance accumulated before reaching the state-action pair $(s,a,h)$. The conditioning on $s_h$ reflects that this quantity is only meaningful along trajectories that actually visit $(s,h)$. This captures the following phenomenon: to obtain sufficiently accurate estimates at $(s,a,h)$ and avoid regret from under-sampling, the agent may need to visit earlier states more frequently than would be necessary. This additional sampling effort is encoded in the conditional variance, making it central to understanding and bounding the regret.

---

> > ### Comment · Reviewer_hoFY · 2025-08-04
> >
> > I find the rebuttal satisfactory and will keep my score. Thank you for the response!

---

### Official Review · Reviewer_gPCR · 2025-07-02

**Clarity:** 2
**Significance:** 3
**Originality:** 3
**Rating:** 5
**Confidence:** 4

**Summary:**

The paper addresses regret bounds of tabular MDPs. They focus on providing gap-dependent regret bounds that are also variance-aware. They study the dependence of regret on the maximum conditional total variance (conditioned on the states) instead of the maximum unconditional total variance studied in existing literature. They show that the existing Monotonic Value Propagation (MVP) algorithm enjoys improved regret bounds. A lower bound on the gap-dependent regret is also provided.

**Questions:**

- Why did the authors choose to present only the informal version of the regret bound in the main paper and leave the formal theorem to the appendix? The formal version of the lower bound is particularly different than the informal version presented in Theorem 3. I strongly recommend including the complete technical theorem in the main paper.

- While the upper bound proof of the regret is well written, it was challenging to follow the presentation of the lower bound, due to the following reasons.

1. A lot of details (including assumptions) are missing from the informal version of the lower bound theorem (Theorem 3).

2. Instead of giving a summarized, intuitive idea of the proof in the proof sketch, a simplified instance is discussed, which does not give much insight into how the instance can be generalized.

3. Even though I am well aware of the multi-armed bandits lower bound proof (Lai & Robbins, 1985) utilized in appendix C, it was difficult to see how to it utilized to get a lower bound here, I request authors to expand and provide more detailed and easy to follow proof of the lower bound in the appendix.


- The abstract includes the mathematical expression for regret bounds. Mathematical expressions are not displayed properly when the abstract is shown outside the PDFs (for e.g., openreview website). Hence, the authors might want to consider rewriting the abstract with fewer (or none) mathematical expressions to improve readability.

- As evident in the questions, my primary concern with the paper is with the writing of Section 5 and the corresponding Appendix C; the score given currently is a reflection of that.

**Ethical Concerns:**

["NO or VERY MINOR ethics concerns only"]

**Final Justification:**

The authors have addressed my questions adequately. I have updated my score accordingly. The overall consensus of the reviewer is positive, and I expect the authors to incorporate all the changes discussed during the rebuttal phase.

**Limitations:**

Yes

**Quality:**

3

**Strengths And Weaknesses:**

**Strengths**
- The paper studies the dependence of regret on the maximum conditional total variance instead of the maximum unconditional total variance studied in existing literature, and shows how it allows a better regret bound for an existing algorithm (Monotonic Value Propagation).
- The introduction of the conditional variance allows improvement in dependence on $H$.
- The techniques used in this paper can be used to improve regret bounds in a variety of RL settings.


**Weaknesses**
- The clarity and structure of the paper can be improved. The paper only includes an informal version of theorems (upper bound and lower bound theorems. This is acceptable if the primary contribution of a paper is algorithmic or experimental. The primary contributions of this paper are regret bounds (upper / lower bounds), so it would be a better idea to include complete formal theorems in the paper. If space is a concern, authors can move the MVP algorithm to the appendix, as the algorithm is not a contribution of this work.

---

> ### Author Rebuttal · Authors · 2025-07-30
>
> > Why did the authors choose to present only the informal version of the regret bound in the main paper and leave the formal theorem to the appendix? The formal version of the lower bound is particularly different than the informal version presented in Theorem 3. I strongly recommend including the complete technical theorem in the main paper.
>
> > The abstract includes the mathematical expression for regret bounds. Mathematical expressions are not displayed properly when the abstract is shown outside the PDFs (for e.g., openreview website). Hence, the authors might want to consider rewriting the abstract with fewer (or none) mathematical expressions to improve readability.
>
> We would like to thank the reviewer for their helpful suggestions regarding the presentation of our main theorem and the rendering of the abstract. These issues will be addressed in the camera-ready version, should the paper be accepted.
>
> > While the upper bound proof of the regret is well written, it was challenging to follow the presentation of the lower bound, due to the following reasons.
>
> We apologize for any confusion caused by the proof of the lower bound. We are happy to clarify any specific concerns. If any part of the proof remains unclear, please indicate the exact step or argument that you find troubling, and we will provide a detailed explanation.
>
> >> A lot of details (including assumptions) are missing from the informal version of the lower bound theorem (Theorem 3).
>
> We chose to present an informal statement of the lower-bound theorem in the main text to balance precision and clarity. The formal version, which includes all technical details and assumptions, is available in the Appendix for interested readers.
>
> >> Instead of giving a summarized, intuitive idea of the proof in the proof sketch, a simplified instance is discussed, which does not give much insight into how the instance can be generalized.
>
> The example in the proof sketch captures the key behavior of the MDP used in the formal lower-bound proof: the agent starts in the main state $\texttt A$, transitions once with high probability to an action state $\texttt B$, and then makes a decision. This structure generalizes to arbitrary $S,A,H$ by introducing more action states, as detailed in the Appendix.
>
> >> Even though I am well aware of the multi-armed bandits lower bound proof (Lai & Robbins, 1985) utilized in appendix C, it was difficult to see how to it utilized to get a lower bound here, I request authors to expand and provide more detailed and easy to follow proof of the lower bound in the appendix.
>
> Finally, the lower bound for multi-armed bandits is used to bound the regret incurred in each action state. Summing these contributions yields a valid lower bound for the regret in the MDP setting.

---

> > ### Comment · Reviewer_gPCR · 2025-08-05
> > **Response to the Rebuttal**
> >
> > I thank the authors for their response. I have updated my score to reflect the responses. If accepted, I recommend that the author make the suggested changes in the abstract. If authors choose to only present informal theorem results in the main body of the paper, I  recommend reducing the technical gap (lack of details, especially assumptions) between the formal theorem and the informal version in the main body of the paper. Thank you!
> >
> > Reviewer gPCR

---

### Official Review · Reviewer_RgUt · 2025-07-03

**Clarity:** 4
**Significance:** 3
**Originality:** 3
**Rating:** 5
**Confidence:** 3

**Summary:**

This paper presents a new theoretical analysis of the Monotonic Value Propagation (MVP) algorithm, establishing the first variance-aware, gap-dependent regret bound using a newly introduced metric: the maximum conditional total variance. The analysis achieves a state-of-the-art worst-case horizon dependence of H^2 in the regret bound, improving over prior work. The authors also prove a matching lower bound, showing that no algorithm can significantly outperform MVP in this setting. Key technical contributions include a novel reweighting analysis of suboptimal gaps and a refined clipping strategy for optimal actions.

**Questions:**

- The favorable bounds of the MVP algorithm shown in this work is very interesting, however, does this tight bound mean the MVP algorithm is very close to an optimal algorithm? Do you think we should still try to develop newer RL algorithms, or just focus on adapting existing ones into practical problems?
- Is there any empirical evidence showing that MVP works much better than alternative RL algorithms? (perhaps even in some special MDPs)
- How easily will the analysis made in the paper be extended to other more general MDP settings?
- Does these theoretical results bring insight on how to better adapt MVP (or similar model-based algorithms)  to settings with function approximation or large state spaces?

**Ethical Concerns:**

["NO or VERY MINOR ethics concerns only"]

**Final Justification:**

The authors have answered my questions in the rebuttal. I believe further empirical results and analysis in the function approximator settings will enhance the paper even more, but I understand that these results are non-trivial and are important future directions.

Given the significant theoretical contribution already presented in this paper, I keep my final score as 5 (accept).

**Limitations:**

Yes

**Quality:**

4

**Strengths And Weaknesses:**

Strengths

**Quality**
- Overall, the paper is very well written.
- The arguments are supported by cleanly organized proofs.

**Clarity**
- The paper is written clearly, terms and concepts are clearly explained or defined.
- Concepts such as suboptimality gaps, conditional variance, and the role of the MVP algorithm are well-motivated and easy to follow for readers with background in RL theory.

**Significance**
- The paper studies the important question of "What is the tightest problem-dependent regret while considering both variance and gap?". Obtaining such a bound can be helpful for the further adaptation of new and potentially better RL algorithms such as MVP.
- The theoretical results are very interesting, providing a tight bound on MVP can be a significant result
- The tight lower bound further strengthens the impact by showing near-optimality.
- The new techniques discussed in the paper can be useful for other theoretical works.


**Originality**
- The theoretical results are new, the new techniques that are used to prove the results are also novel

Weaknesses

**Significance**
- While the theoretical results are impressive, the paper does not include any empirical evaluation, whether on small toy problems or other more complex settings. It is unclear how these theoretical improvements translate to practical reinforcement learning problems. But I understand the main goal of this paper is to provide theoretical contributions.

---

> ### Author Rebuttal · Authors · 2025-07-31
>
> > The favorable bounds of the MVP algorithm shown in this work is very interesting, however, does this tight bound mean the MVP algorithm is very close to an optimal algorithm? Do you think we should still try to develop newer RL algorithms, or just focus on adapting existing ones into practical problems?
>
> We believe there is still significant room for developing new RL algorithms. A result by Simchowitz et al. shows that a regret term of order $S/\Delta_{\min}$ is unavoidable for any optimistic algorithm—this limitation is reflected in the second term of our regret bound. In contrast, Xu et al. propose the AMB algorithm, which reduces this term to $|\mathcal Z_{\mathrm{mul}}|/\Delta_{\min}$ in a variance-independent setting, where $|\mathcal{Z}_{\mathrm{mul}}|$ is a more refined structural quantity (equal to zero when every state has a unique optimal action). This comparison suggests that there may be unexplored algorithmic directions that combine the variance-awareness of MVP with the structural adaptivity of AMB. Hence, we are optimistic about future algorithmic advances beyond MVP.
>
> > Is there any empirical evidence showing that MVP works much better than alternative RL algorithms? (perhaps even in some special MDPs)
>
> Empirical comparisons between MVP and other RL algorithms are nontrivial, especially in finite-horizon tabular MDPs. The performance is often sensitive to constant factors and logarithmic terms, which makes isolating the impact of the theoretical improvements quite difficult. At this stage, we do not have conclusive empirical results supporting a performance advantage for MVP. A rigorous experimental study, particularly on carefully designed MDPs that highlight the role of variance, is an important direction for future work.
>
> > How easily will the analysis made in the paper be extended to other more general MDP settings?
>
> We believe the techniques in our analysis can be extended to other gap-dependent settings without significant difficulty. The key challenge typically lies in identifying appropriate weighting schemes that lead to tight bounds. Once those are identified, our framework for incorporating conditional variance could likely be reused or adapted with modest effort.
>
> > Does these theoretical results bring insight on how to better adapt MVP (or similar model-based algorithms) to settings with function approximation or large state spaces?
>
> To our knowledge, there has not yet been an attempt to generalize the analysis of MVP—or other similar model-based algorithms—to the function approximation setting. We view this as an important and open problem. While our theoretical results provide some insight into how variance affects regret in the tabular case, extending these insights to large or continuous state spaces will likely require new techniques. We consider this a promising avenue for future research.

---

### Official Review · Reviewer_unm4 · 2025-07-03

**Clarity:** 4
**Significance:** 3
**Originality:** 4
**Rating:** 4
**Confidence:** 4

**Summary:**

The authors introduce the notion of conditional total variance into a new instance dependent (bounds depend on the gap of the MDPand total conditional variance quantity) performance bounds of the MVP algorithm of Zhang et al 2024. The paper also has a matching lower bound, which is the first in this style of performance bounds.

**Questions:**

Can the authors comment if there are other lower instance dependent lower bounds which may not be "variance-aware" ? How does their lower bounds compare with those?

Do the authors make any changes to MVP algorithm ? or one should consider this paper as proving properties for the MVP algorithm?

**Ethical Concerns:**

["NO or VERY MINOR ethics concerns only"]

**Final Justification:**

I retain my score.
I was satisfied with the author's rebuttal. While this paper presents theoretical results that may be of value, their impact may be somewhat limited due to the limited empirical guarantees.

**Limitations:**

lack of empirical validation, potentially limited impact

**Quality:**

3

**Strengths And Weaknesses:**

S:
1)The paper is extremely well written and easy to follow. Clarity and organization is very good.
2)The lower bounds are a definitely an exciting addition to the literature.
3)The proof techniques seem novel to me.

W:
1)A detailed discussion about the significance of conditional variance is needed. e.g., what are some cases where conditional variance is significantly small? (better still , small experiments to depict this intuition)
2) Why is this a tighter or more appropriate notion than unconditional total variance (Def 1).
3) How does the MVP algorithm behave with small \Delta_min.

---

> ### Author Rebuttal · Authors · 2025-07-31
>
> > A detailed discussion about the significance of conditional variance is needed. e.g., what are some cases where conditional variance is significantly small? (better still , small experiments to depict this intuition)
>
> > Why is this a tighter or more appropriate notion than unconditional total variance (Def 1).
>
> Thank you for raising this point about the distinction between conditional and unconditional variance. In Appendix C, we construct an MDP instance that matches our lower bound, where the unconditional variance is significantly smaller than the conditional variance. This demonstrates that the unconditional variance fails to capture the complexity relevant to gap-dependent regret, making it inadequate for deriving tight lower bounds in such settings. We will clarify this distinction more explicitly in the camera-ready version, should the paper be accepted.
>
> That said, empirical investigation of MVP is not straightforward. Currently, we lack conclusive evidence that changes in performance can be attributed specifically to differences in (un)conditional variance, as other factors—such as suboptimality gaps or constant scaling—may dominate. As a result, designing meaningful experiments to illustrate this point remains an open challenge.
>
> > How does the MVP algorithm behave with small \Delta_min.
>
> We suspect you are referring to the second term $\sum_{\Delta_h(s,a)=0}\frac{H^2\wedge \texttt V\texttt a\texttt r_c^*}{\Delta_\min}$ in our bound. While this term does not appear in our lower bound, Simchowitz et al. (2021) established that a regret term of order $\frac{S}{\Delta_{\min}}$ is unavoidable for any optimistic algorithm. Our $\Delta_{\min}$-dependent term reflects this intrinsic limitation. For a more in-depth discussion, we refer you to the aforementioned work.
>
> > Can the authors comment if there are other lower instance dependent lower bounds which may not be "variance-aware" ? How does their lower bounds compare with those?
>
> To the best of our knowledge, there are no other instance-dependent lower bounds that incorporate both suboptimality gaps and variance information. However, there are bounds that focus solely on the suboptimality gaps. As mentioned earlier, Simchowitz et al. derive a lower bound of $O(\sum_{\Delta_h(s,a)>0} \frac{1}{\Delta_h(s,a)} + \frac{S}{\Delta_{\min}})$ for optimistic algorithms. Xu et al. later generalized this bound to cover all algorithms when $S$ is logarithmically small. Our work extends this line of research by introducing conditional variance as an additional key factor.
>
> > Do the authors make any changes to MVP algorithm ? or one should consider this paper as proving properties for the MVP algorithm?
>
> The MVP algorithm analyzed in our paper is a slightly modified version of the original algorithm proposed by Zhang et al., which includes a doubling update trick. We believe this trick is intended to reduce update costs in the minimax setting and is not essential for the gap-dependent analysis considered here. Therefore, we omit it in our version. Relevant discussion can be found in the footnote on page 6.

---

> > ### Comment · Reviewer_unm4 · 2025-08-05
> >
> > Thanks for the rebuttal.
> > I am satisfied with the author's rebuttal. I retain my score

---

### Note · Authors · 2025-08-16

We sincerely thank the reviewers again for their valuable feedback. We would like to take this opportunity to address two commonly raised concerns.

1. On the definition of conditional variance.
We chose to define conditional variance in a way that conditions on a later state because it intuitively measures the difficulty of learning the $V$ and $Q$ values at a given state. This captures the fact that inefficient sampling at one state can affect estimation at all previous states. Moreover, Theorem 3 shows that conditional variance lower-bounds regret even when the unconditional variance is small, which justifies its role in our regret bound.
2. On the $\Delta_\min^{-1}$-dependent term.
Simchowitz et al. have shown that such a term is unavoidable for any optimistic algorithm, due to over-exploration. Our second term directly reflects this inherent limitation of the MVP algorithm. We hope future work will explore non-optimistic algorithms that can mitigate or eliminate this term.

We believe these clarifications address the main concerns raised in the reviews.

---

### Decision · Program_Chairs · 2025-09-17

**Decision:**

Accept (poster)

**Comment:**

This work derives a tighter instance-dependent regret bound for the MVP algorithm in the episodic, tabular MDP setting. Unlike most prior works covering this topic, the authors focus on tightening the variance terms in the instance dependent bounds. The regret upper bounds are supplemented by a lower bound which matches the upper bound term containing the sub-optimal states, which is quite standard for optimism based algorithms.

All reviewers are happy with the theoretical contributions of the paper, find that the techniques used may be of independent interest and think that the notion of conditional variance is interesting. Reviewers are also happy with the matching lower bound. Most reviewers were also very happy with the presentation of the paper and the quality of the writing. The only minor concerns were about lack of empirical evaluation and questionable practical impact of the work. Given that this is a purely theoretical paper that explores new types of regret bounds for existing algorithms I think the work stands on its own without the empirical evaluation and recommend acceptance.